# Selective cargo and membrane recognition by SNX17 regulates its interaction with Retriever

Aurora Martín-González [1,4], Iván Méndez-Guzmán[1,4], Maialen Zabala-Zearreta [2], Andrea Quintanilla[1], Arturo García-López [1], Eva Martínez-Lombardía[1], David Albesa-Jové[2,3], Juan Carlos Acosta [1] & María Lucas [1✉]

## Abstract

The Retriever complex recycles a wide range of transmembrane proteins from endosomes to the plasma membrane. The cargo adapter protein SNX17 has been implicated in recruiting the Retriever complex to endosomal membranes, yet the details of this interaction have remained elusive. Through biophysical and structural model-guided mutagenesis studies with recombinant proteins and liposomes, we have gained a deeper understanding of this process. Here, we demonstrate a direct interaction between SNX17 and Retriever, specifically between the C-terminal region of SNX17 and the interface of the Retriever subunits VPS35L and VPS26C. This interaction is enhanced upon the binding of SNX17 to its cargo in solution, due to the disruption of an intramolecular auto-inhibitory interaction between the C-terminal region of SNX17 and the cargo binding pocket. In addition, SNX17 binding to membranes containing phosphatidylinositol-3-phosphate also promotes Retriever recruitment in a cargo-independent manner. Therefore, this work provides evidence of the dual activation mechanisms by which SNX17 modulates Retriever recruitment to the proximity of cargo and membranes, offering significant insights into the regulatory mechanisms of protein recycling at endosomes.

Keywords Intracellular Trafficking; Endosomes; Retriever; Sorting Nexins; SNX17
Subject Category Membranes & Trafficking

## Introduction

The abundance and distribution of integral membrane proteins on the cell surface regulate a wide range of cellular functions, including signaling, adhesion, migration, and nutrient transport, all essential for maintaining cellular homeostasis. In addition to the secretory pathway, the endolysosomal network plays a crucial role in regulating protein composition at the plasma membrane by sorting internalized proteins (called cargo), either for lysosomal degradation or for recycling back to the plasma membrane or trans-Golgi network (Cullen and Steinberg, 2018; McNally and Cullen, 2018). Disruptions in this process are linked to various human diseases, including Alzheimer's and Parkinson's (McMillan et al, 2017; Small et al, 2017; Schreij et al, 2016).

Two main sorting pathways have been identified for recycling cargo proteins from endosomes to the plasma membrane: the Retromer complex, in combination with SNX27 and the ESCPE-1 complex, and the Retriever complex, in combination with SNX17 and the CCC complex (Yong et al, 2023; Simonetti et al, 2023; Gopaldass et al, 2024). The WASH complex also acts in both pathways by inducing the formation of branched actin networks (Simonetti and Cullen, 2019).

The well-characterized Retromer complex, composed of VPS26, VPS29, and VPS35 subunits (Seaman et al, 1998, 1997), functions with sorting nexins in cargo selection and promotes membrane deformation to generate tubulovesicles, which transport sorted cargos to either the plasma membrane or the TGN (Hierro et al, 2007; Lucas et al, 2016; Kovtun et al, 2018; Leneva et al, 2021; Carosi et al, 2023). The Retriever complex, identified seven years ago, is a stable complex, composed of VPS26C, VPS29, and VPS35L subunits, highly conserved in vertebrates and expressed in nearly all human cell types (Mallam and Marcotte, 2017; McNally et al, 2017; Gershlick and Lucas, 2017; Rabouille, 2017; Wang et al, 2018; Chen et al, 2019). Retriever recycles key proteins, including β1-integrin, LRP1 (McNally et al, 2017) and LDLR (Vos et al, 2023), and is essential for fetal development in mice (Kato et al, 2020). Recent studies have also shown that the SNX17-Retriever recycling pathway regulates synaptic function and plasticity (Rivero-Ríos et al, 2023) and is associated with neurodevelopmental disorders (Kato et al, 2020; Beetz et al, 2020; Otsuji et al, 2023).

Retriever acts in collaboration with the CCC complex and DENND10 as part of a large multisubunit assembly known as the Commander complex (Wan et al, 2015; Healy et al, 2023; Boesch et al, 2023). The CCC complex comprises 12 subunits: CCDC22, CCDC93, and ten members of the COMMD family. This complex is involved in the recycling of various cargos, such as ATP7A, ATP7B, GLUT1, LDLR, and NOTCH2 (Bartuzi et al, 2016; Singla et al, 2019, 2021). Recent studies, which integrate data from cryo-

[1]Instituto de Biomedicina y Biotecnología de Cantabria (IBBTEC), Universidad de Cantabria-CSIC, Santander 39011, Spain. [2]Instituto Biofisika (CSIC, UPV/EHU), Fundación Biofísica Bizkaia/Biofisika Bizkaia Fundazioa (FBB) and Departamento de Bioquímica y Biología Molecular, University of the Basque Country, 48940 Leioa, Spain. [3]Ikerbasque, Basque Foundation for Science, 48013 Bilbao, Spain. [4]These authors contributed equally: Aurora Martín-González, Iván Méndez-Guzmán. ✉E-mail: maria.lucas@unican.es

electron microscopy (cryo-EM), X-ray crystallography, AlphaFold2 (AF2) modeling, and mutational analysis have provided significant insights into the structural organization of the Commander complex (Healy et al, 2023; Boesch et al, 2023; Laulumaa et al, 2024).

Sorting nexin proteins SNX17 and SNX31 have been implicated in recruiting the Retriever complex to endosomal membranes (McNally et al, 2017; Rivero-Ríos et al, 2023). Both bind endosomal membranes through a PX domain that specifically recognizes phosphatidylinositol-3-phosphate (PI3P) (Chandra et al, 2019), and engage with the cytosolic tail of transmembrane proteins with the sorting motifs NPxY or NxxY through a FERM-like domain (Ghai et al, 2013). SNX17 has a predicted disordered region at its C-terminal end (CT), which has been proposed to interact with Retriever (McNally et al, 2017), though the exact recruitment mechanism remains unclear. Elucidating this process is crucial for fully understanding the molecular mechanism underlying the function of the Retriever-SNX17-CCC pathway.

In this study, we present a model that clarifies the association of SNX17 with Retriever based on small-angle X-ray scattering (SAXS) data, AF2 modeling, biophysical assays, and mutagenesis studies. We demonstrate that the CT region of SNX17 is directly involved in an autoinhibition mechanism, which is relieved upon either cargo binding or membrane association, enabling the SNX17 C-terminal region to bind the VPS35L-VPS26C interface.

# Results and discussion

## The architecture of the Retriever complex in solution

Retriever is a heterotrimer composed of the proteins VPS26C, VPS29, and VPS35L (Fig. 1A). While recent cryo-EM studies have provided structural insights, little is known about the molecular properties, flexibility, and stability of Retriever and its subunits in solution. To address this, the Retriever complex was purified from insect cells, and individual subunits VPS26C and VPS29 were expressed in bacteria (Appendix Fig. S1). Attempts to isolate VPS35L alone were unsuccessful, suggesting that it requires association with the other subunits for proper folding.

Purification experiments of the Retriever complex with constructs of different VPS35L lengths (Fig. EV1A,B) revealed that VPS35L must interact with VPS26C or VPS29 to maintain structural integrity in solution. VPS26C binds the N-terminal region of VPS35L (residues 110–598), while VPS29 binds VPS35L through interactions with both the N-terminal (residues 1–110) and C-terminal (residues 599–963) regions, a dual binding mechanism confirmed by cryo-EM (Fig. EV1C,D) (Healy et al, 2023; Boesch et al, 2023; Laulumaa et al, 2024).

Further analysis using size exclusion chromatography coupled with small-angle X-ray scattering (SEC-SAXS) highlights the flexible nature of VPS26C, in contrast to the overall compact structure of the Retriever complex (Fig. 1B–I; Appendix Fig. S2; Appendix Table S1). This flexibility in VPS26C may facilitate its interaction with VPS35L for Retriever complex formation and potentially enable Retriever interactions with other partners. Structural modeling using AF2 and cryo-EM data aligned well with the SAXS results (Fig. 1E,I), confirming that these models accurately represent the conformation of VPS26C and Retriever in solution.

# SNX17 interaction with cargo triggers Retriever recruitment

Previous studies suggested an interaction between SNX17 and Retriever through VPS26C, based on co-immunoprecipitation assays (McNally et al, 2017). However, subsequent research failed to replicate this interaction (Healy et al, 2022). Given the similarities with Retromer, whose interaction with SNX3 depends on cargo presence (Lucas et al, 2016), we explored whether SNX17-Retriever interaction is also cargo-dependent.

First, we examined the binding affinity between $SNX17_{FERM-CT}$ (amino acids 109-470) and several cargos, including low-density lipoprotein receptor-related protein 1 (LRP1), amyloid precursor protein (APP), β1 integrin (ITGB1), and human papillomavirus type 16 (HPV16) capsid protein L2 (L2), all previously identified as SNX17 cargos (Farfán et al, 2013; Lee et al, 2008; Steinberg et al, 2012; Bergant Marušič et al, 2012) (Fig. 2A). Fluorescence anisotropy assays with 5-FAM-labeled peptides covering the FERM (NPxY) binding motif (Fig. 2B) showed that $SNX17_{FERM-CT}$ exhibited high affinity for L2 ($K_D = 0.6$ μM) and LRP1 ($K_D = 2.2$ μM), moderate affinity for APP ($K_D = 19$ μM), and variable affinity for ITGB1, with the membrane-proximal site ($K_D = 72$ μM) showing higher affinity than the distal site ($K_D = 125$ μM) (Fig. 2C). The central region of L2 displayed a remarkably high affinity for SNX17 and could outcompete the physiological cargos of SNX17.

To investigate the contribution of cargo to the SNX17-Retriever interaction, we employed pull-down assays. MBP-SNX17 was incubated with Retriever in the presence or absence of cargo proteins, including $LRP1_{ICD}$ (residues 4444–4544), $APP_{ICD}$ (residues 724–770), $ITGB1_{ICD}$ (residues 752–798) and $L2_{FBR}$ (the FERM binding region, residues 239–268) (Fig. 2A). In the absence of cargo, only a modest amount of Retriever was retained, but it increased significantly with the addition of $LRP1_{ICD}$ or $L2_{FBR}$ (Fig. 2D). No significant interaction was detected with $APP_{ICD}$ or $ITGB1_{ICD}$, likely due to their low affinity for SNX17 (Fig. 2C). This enhanced binding of Retriever to SNX17 was attributed to the interaction of the cargos with SNX17, as no direct interaction between Retriever and these cargos was observed in GST pull-downs (Fig. EV2A). Furthermore, mutating the NPxY motif in LRP1 and L2 abolished SNX17 binding, and no significant increase in Retriever retention was observed in MBP-SNX17 pull-downs with these mutants (Fig. EV2B). Due to their high binding affinity to SNX17, LRP1 and L2 were chosen for further in vitro studies of the cargo-SNX17-Retriever complexes. Despite L2 being a viral cargo, it is predicted to bind to the same SNX17 pocket as cellular cargos (Bergant Marušič et al, 2012; Bergant et al, 2017).

The L2-SNX17 interaction is essential for HPV infection, being necessary for optimal capsid disassembly and facilitating the viral escape from late endosomes (Bergant Marušič et al, 2012; Bergant et al, 2017). Two studies have shown that siRNA-mediated knockdown of Retriever reduces the infection efficiency of HPV16 pseudovirions (McNally et al, 2017; Pim et al, 2021). In addition, Retriever has been observed to colocalize with L2 in HeLa cells infected with HPV16 pseudovirions (Pim et al, 2021). To investigate the L2-mediated association of SNX17 with Retriever, we overexpressed the $L2_{FBR}$ fragment and full-length L2 in HEK293T cells (Appendix Fig. S3). Immunoprecipitation showed that SNX17 and VPS35L associate with L2, with the FBR domain being sufficient for SNX17 binding. Mutation of the $^{254}NPxY^{257}$ motif in L2 blocked SNX17 binding to the FBR domain but not to full-length L2,

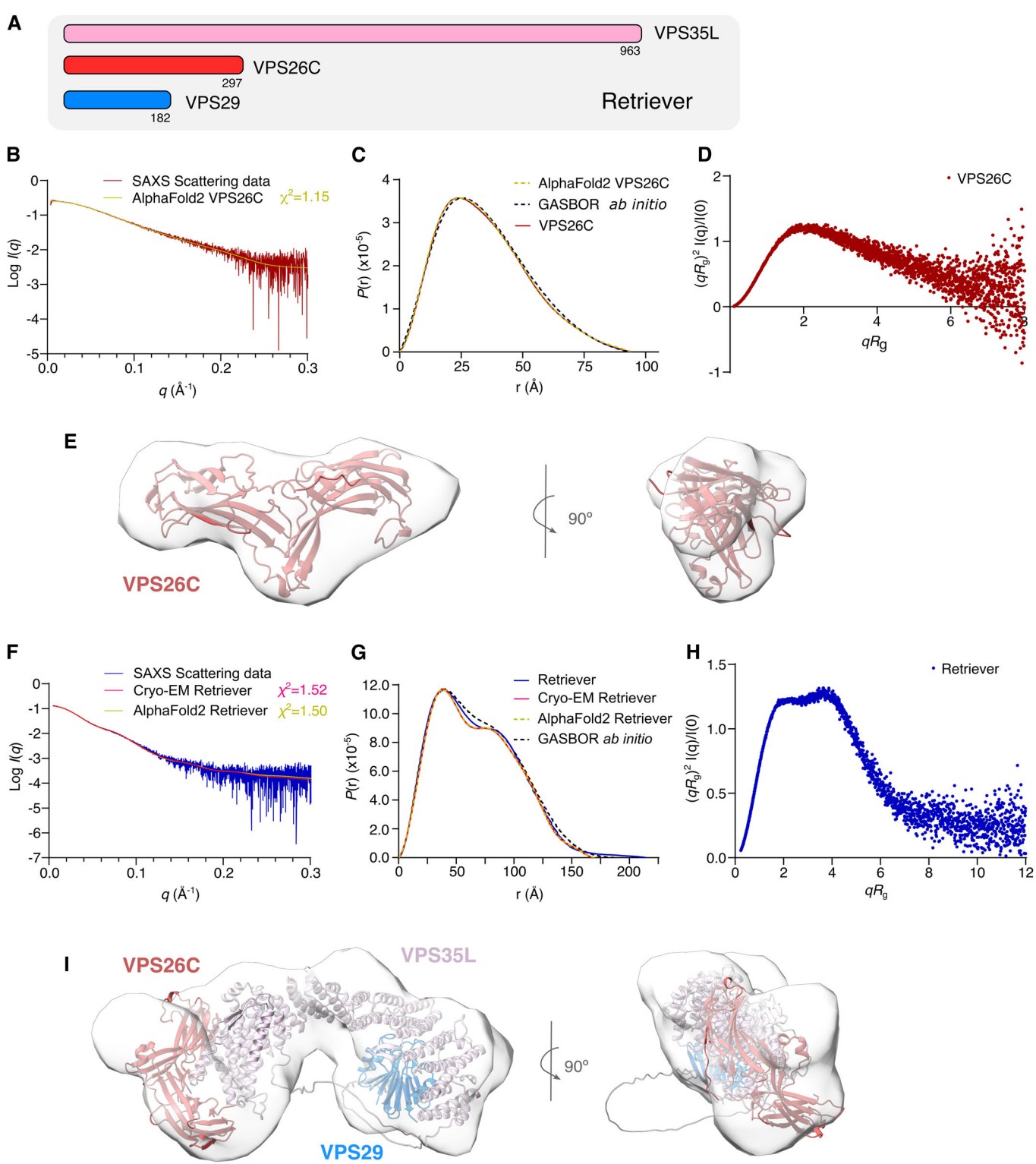

suggesting additional SNX17 binding sites. VPS35L retention by both wild-type and mutant full-length L2 suggests Retriever association with L2 outside of the FBR, possibly directly or via SNX17. Further research is needed to clarify how L2 hijacks Retriever.

We also examined the effect of ionic strength on cargo-mediated SNX17-Retriever interaction. At low salt concentrations (50 mM NaCl), a strong interaction occurred even without cargo, likely due to non-physiological binding. Increasing salt to physiological levels significantly reduced this interaction in the absence of cargo (Fig. EV2C). Subsequent assays were conducted at 200-300 mM NaCl to better mimic physiological conditions.

**Figure 1. SAXS analysis of the Retriever complex and its comprising subunit VPS26C.**

(A) Scheme of the subunits comprising the Retriever complex. (B) SEC-SAXS scattering profile of VPS26C. The theoretical scattering curve for the VPS26C AF2 model (UniProt O14972) was calculated and compared to the experimental scattering curve using CRYSOL ($\chi^2 = 1.15$). (C) Pair-distance distribution functions, $P(r)$, of VPS26C based on SAXS experimental data (solid red line), the AF2 model (green dashed line), and the GASBOR dummy atom model (black dashed line). (D) Dimensionless Kratky plot of VPS26C. (E) Overlay of the AF2 model of VPS26C within the ab initio SAXS envelope calculated using the GASBOR algorithm (gray). The envelope and subsequent fitting were generated using UCSF ChimeraX. (F) SEC-SAXS scattering profile of Retriever with CRYSOL fit for the AF2 Retriever model (Model Archive ID: ma-3cag5, $\chi^2 = 1.50$), and the Retriever cryo-EM structure (PDB 8SYN, with the missing loops modeled using AF2, $\chi^2 = 1.52$). (G) Pair-distance distribution functions of Retriever were calculated from the SAXS experimental data (solid blue line), the cryo-EM structure (solid pink line), the AF2 model (green dashed line), and the GASBOR dummy atom model (black dashed line). (H) Dimensionless Kratky plot of the Retriever complex. (I) Overlay of the AF2 model of Retriever (Model Archive ID: ma-3cag5) within the ab initio SAXS envelope calculated with the GASBOR algorithm (gray). Source data are available online for this figure.

## The C-terminal end of SNX17 contacts the VPS35L/VPS26C interface

To map the interaction region between SNX17 and Retriever, we modeled various complex configurations with AF2-multimer: Retriever-SNX17 (Fig. 3A; Appendix Fig. S4A) and VPS26C:VPS35L$_{110-598}$ in complex with either the last 18 residues of SNX17 (SNX17$_{CT-18}$) (Fig. 3B and Appendix Fig. S4B), or with full-length SNX17 and the cargo peptide L2$_{17mer}$ (residues 245–261) (Appendix Fig. S4C,D). In all models, the C-terminal 18 residues of SNX17 consistently bind to a groove at the VPS26C-VPS35L interface, forming several polar and hydrophobic contacts. This interaction is stabilized by a VPS35L β-hairpin, termed the "hinge region", which includes salt bridges between specific VPS35L and SNX17 residues (Fig. 3B). Notably, this hinge region is absent in the cryo-EM structure of Retriever (Boesch et al, 2023) and displays low pLDDT (predicted local distance difference test) scores in our AF2 models (Appendix Fig. S4A–C), suggesting flexibility that likely facilitates SNX17 insertion. The position of the last 12 residues of SNX17 at the VPS26C-VPS35L interface is supported by several observations: high pLDDT scores (pLDDT >90 for the last four residues and 73 < pLDDT < 90 for residues 459–465), PAE (predicted aligned error) plots indicating high-confidence alignment with Retriever subunits (Appendix Fig. S4B), and the evolutionary conservation of these residues (Fig. 3B; Appendix Figs. S5–7). Pull-down assays confirmed that the FERM-CT region is crucial for Retriever recruitment (Fig. EV3A).

In the AF2-multimer models, SNX17 C-terminal residue L470 establishes a hydrophobic interaction with VPS35L W280, while its carboxyl group forms a salt bridge with VPS35L residue R248 (Fig. 3B). Truncation of the last four residues (D467X) and the substitution of the terminal leucine with glycine (L470G) (McNally et al, 2017) disrupt Retriever binding (Fig. 3C), validating the predicted interaction. These substitutions did not significantly alter the protein structure, as shown by circular dichroism (CD) (Appendix Fig. S8), nor disrupt cargo binding (Fig. EV3B). Binding affinity assays with SNX17$_{CT-18}$ confirmed a $K_D$ of 5.7 μM for Retriever (Fig. 3D). Additionally, the Retriever-binding region was mapped using different Retriever complex configurations. SNX17$_{CT-18}$ displayed a comparable affinity for full-length Retriever and VPS26C-VPS35L$_{110-598}$ ($K_D = 5.3$ μM), no discernible binding to VPS26C or VPS29, and only a weak interaction with VPS35L-VPS29. Point mutations in VPS35L validated the interaction cavity, as mutations R248E + W280D, and K157E + R161E reduced or eliminated binding. Mutation R248E + W280D was designed to disrupt the interaction of VPS35L with SNX17 residue L470, and mutation K157E + R161E was intended to disrupt the salt bridges between K157 and R161 within the VPS35L hinge region and their counterparts in SNX17 (residues E468 and D469, respectively).

An additional SNX17-VPS26C interface is predicted in the SNX17:VPS26C:VPS35L$_{110-598}$:L2$_{17mer}$ complex, involving the VPS26C surface at the apex of the two β-sandwich domains, and the SNX17 FERM F1 and F2 regions (Appendix Fig. S4C). This interaction is observed in eight of ten models generated, with contact residues evolutionarily conserved across human orthologs, including the early Metazoa *Nematostella vectensis* (Appendix Fig. S4D). However, no direct interaction between SNX17 and VPS26C was detected in pull-down and isothermal titration calorimetry (ITC) assays (Appendix Fig. S9). We suggest that the high affinity of the C-terminal tail of SNX17 for the VPS26C-VPS35L interface entraps Retriever and facilitates the potentially weaker interaction between SNX17 and VPS26C, thereby drawing Retriever towards the membrane.

The interaction of Retriever with SNX17 shares some similarities and fundamental differences when compared to the interaction between Retromer and other sorting nexins. Retromer interaction with SNX3 also involves an unstructured domain, the N-terminal domain, that interacts with the VPS26-VPS35 interface, although additional contacts between SNX3 and VPS26 are necessary for stable binding (Harrison et al, 2014; Lucas et al, 2016; Leneva et al, 2021). Retromer also interacts with SNX27 (Steinberg et al, 2013), where the PDZ domain of SNX27 simultaneously engages with the VPS26 subunit of Retromer and cargo through distinct interfaces, in a cooperative manner, facilitating that SNX27 also recruits both Retromer and cargo simultaneously (Gallon et al, 2014). However, the regulation for this recycling pathway is influenced by cargo phosphorylation, a feature not yet identified in the SNX17-Retriever pathway (Clairfeuille et al, 2016).

## SNX17 autoinhibition mechanism for Retriever binding in the absence of cargo

We then investigated the molecular mechanism controlling SNX17 activation for Retriever binding via cargo interaction. None of the published SNX17 structures (Ghai et al, 2011, 2013; Stiegler et al, 2014) feature the CT region, which is predicted to be disordered. However, the AF2 model of SNX17 (Fig. 4A) shows that this region includes a β-strand and the subsequent α-helix (residues 455–464) located within the cargo binding pocket (Fig. 4B). In this CT region, we identified a partially conserved NxxY FERM binding motif with phenylalanine replacing tyrosine (Fig. 4C), forming a cargo-mimicking region. This motif is conserved across SNX17 orthologs (Appendix Fig. S5), and in their AF2 model, this region binds the cargo binding pocket with high-confidence PAE values (Appendix Fig. S10). Cargo association consistently causes a shift of the CT domain in the AF2 models, leading to a pronounced conformational change in this region,

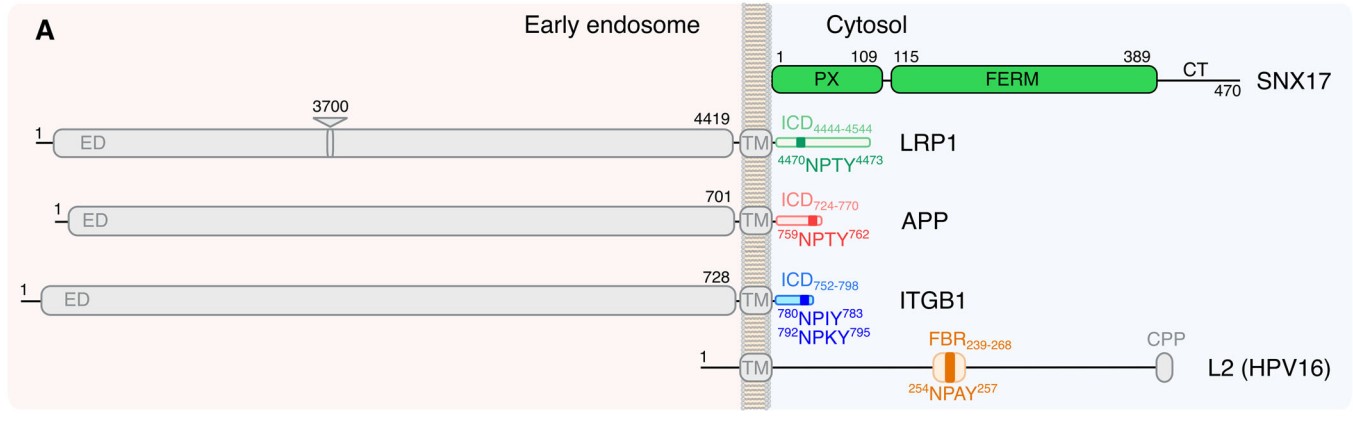

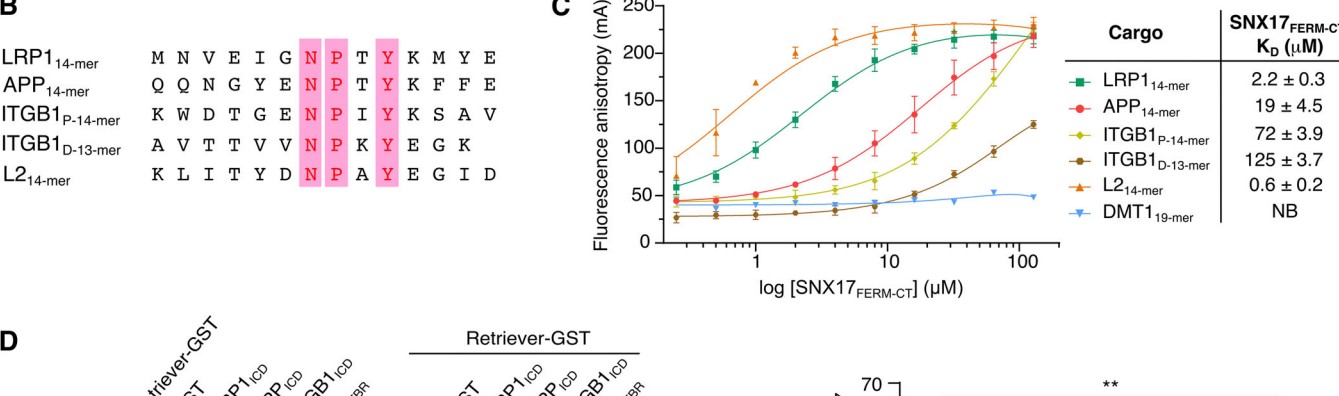

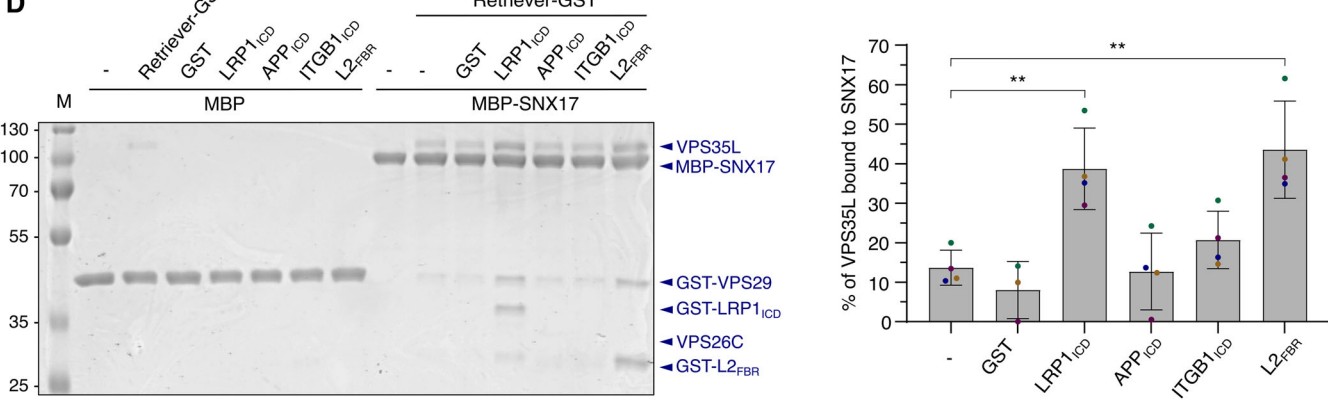

**Figure 2. SNX17 interaction with cargo triggers Retriever recruitment.**

(A) Schematic representation of the domains from SNX17 and the cargos studied in this work. The recycling signaling motif NPxY is highlighted in the diagram. The intracellular domain (ICD) of the physiological cargos and the FERM binding region (FBR) of the L2 protein that were fused with GST for pull-down assays are also highlighted. The length of the proteins is scaled according to their respective number of amino acids, except for LRP1. The depicted triangle in LRP1 corresponds to the missing sequence of 3700 amino acids. CT C-terminal domain, ED extracellular domain, TM transmembrane domain, CPP cell-penetrating peptide. (B) Alignment of the sequences of cargo peptides used in fluorescence anisotropy assays. The conserved NPxY motif is highlighted by pink bars. (C) Fluorescence anisotropy assays were performed to study the interaction of SNX17$_{FERM-CT}$, with the peptides outlined in panel (B). Peptides were labeled with the fluorescent reagent 5-FAM at the N-terminus. The data points on the graph represent the mean ± standard deviation (SD) across three technical replicates. The line represents the fit to the data. The values for the dissociation constants are presented in the table. The Retromer-dependent cargo DMT1 was used as a negative control (Tabuchi et al, 2010). NB no detectable binding. (D) The interaction of the Retriever complex with MBP-SNX17 was evaluated in the presence and absence of the cargos LRP1$_{ICD}$, APP$_{ICD}$, ITGB1$_{ICD}$, and L2$_{FBR}$, each fused with GST, in MBP pull-down assays. Non-fused MBP protein was used as a negative control. Proteins were visualized by Coomassie Blue staining. The right panel shows the quantification of the Retriever binding to SNX17. Quantification was carried out using ImageJ, measuring VPS35L as a representative band of the Retriever complex. The ratio of the VPS35L pull-down band to the MBP-SNX17 band was calculated in each lane, assuming a one-to-one binding stoichiometry. Non-specific binding of VPS35L to MBP was subtracted from the VPS35L band intensities. The results are expressed as mean ± SD (n = 4 biological replicates). Statistical analysis was performed using an unpaired Student's t-test, with cargo vs. without cargo. **p = 0.004. M, protein marker. Source data are available online for this figure.

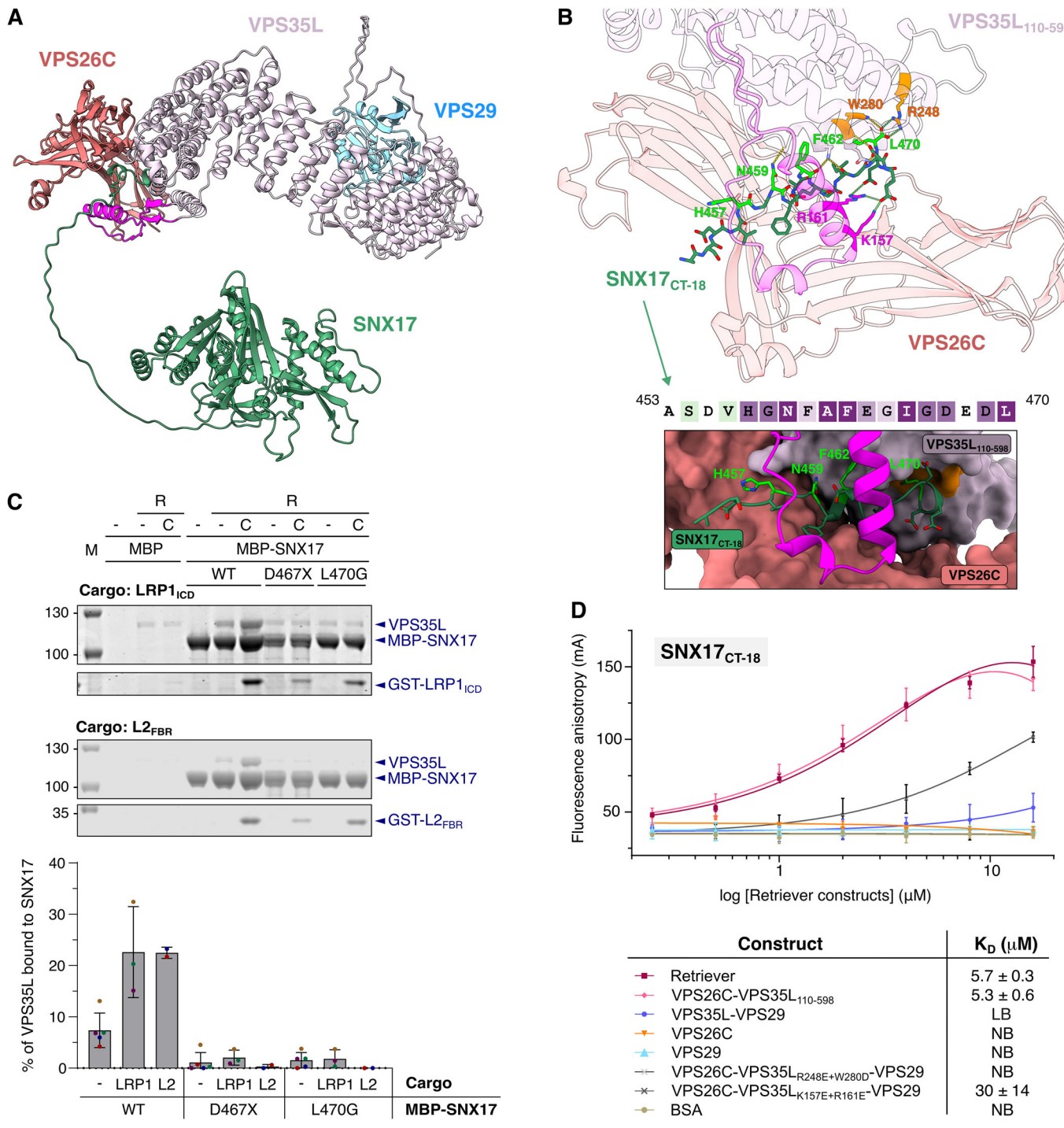

characterized by the emergence of a highly flexible loop comprising the Retriever-binding region at the C-terminus (Fig. 4B; Appendix Fig. S11). Based on these observations, we propose the following autoinhibition mechanism: (i) in the absence of cargo, SNX17 exhibits minimal affinity for Retriever, since the cargo-mimicking region occupies the cargo binding groove, making the overlapping Retriever-binding region poorly accessible; (ii) cargo binding disrupts this inhibitory arrangement, exposing the Retriever-binding region; and (iii) the CT domain, spanning 80 amino acids (residues 390–470), acts

as a dynamic hook to capture Retriever and brings it closer to the endosomal surface (Fig. 4D).

To experimentally validate this proposed autoinhibition mechanism, we introduced specific mutations (Fig. 4A). Mutations W321A and V380D in the SNX17 cargo binding pocket effectively prevent cargo binding. In contrast, the N459A + F462A mutations in the cargo-mimicking region enhanced cargo binding affinity, confirming the cargo-mimicking role (Fig. 4E). This double mutation likely frees the cargo binding pocket, thereby increasing

**Figure 3.    The C-terminal end of SNX17 contacts the VPS35L/VPS26C interface.**

(A) Cartoon representation of the AF2 model of the SNX17:Retriever complex. VPS26C is colored in red, VPS35L in light pink, SNX17 in green, and the hinge region of VPS35L (residues 135–178) is colored in magenta. The confidence estimation for the AF2 model is detailed in Appendix Fig. S4A. (B) Top panel: Detailed view of the SNX17$_{CT-18}$ interaction with the VPS35L-VPS26C interface in the AF2 model of the complex SNX17$_{CT-18}$:VPS26C:VPS35L$_{110-598}$. Proteins are color-coded as in (A). Residues involved in H-bonds (yellow dashed lines), salt bridges (green dashed lines), and the SNX17$_{CT-18}$ chain are shown as sticks. The mutated residues are highlighted with a different color: SNX17 residues with light green, residues of the VPS35L C-terminal binding pocket in orange, and residues of the VPS35L hinge region in magenta. Middle panel: The sequence of SNX17$_{CT-18}$ colored according to sequence conservation calculated with the ConSurf server using a green-through-purple scale, corresponding to variable (grade 1) through conserved (grade 9) positions. Bottom panel: Zoomed-in-view of the binding surface of SNX17$_{CT-18}$. SNX17 residues are shown as cartoon and sticks, and the hinge region of VPS35L (residues 135–178) is in magenta. Mutated residues are colored as in the top panel. (C) Analysis of the interaction between Retriever and SNX17 mutants in the presence and absence of the cargos LRP1 or L2. MBP pull-down assays were performed with wild-type MBP-SNX17 or indicated mutants, Retriever, GST-LRP1$_{ICD}$ or GST-L2$_{FBR}$. Non-fused MBP protein was used as a negative control. Samples were loaded onto an SDS-PAGE gel and stained with Coomassie Blue. Quantification was carried out as detailed in Fig. 2D. The graph represents the mean ± SD of technical replicates (LRP1: $n = 3$; L2: $n = 2$). (D) Fluorescence anisotropy assays measuring direct interaction between 5-FAM-labeled SNX17$_{CT-18}$ peptide and the indicated Retriever constructs to delimit the SNX17 binding region in Retriever. BSA protein was used as a negative control. Data points are the mean ± SD of two technical replicates. Bottom panel: $K_D$ values ± SD calculated using GraphPad Prism, unless too weak to be determined. M protein marker, R Retriever, C cargo, WT wild-type, NB no detectable binding, LB low binding (poor fit quality). Source data are available online for this figure.

the affinity for cargos. On the other hand, the H457A mutation in the cargo-mimicking region had no significant effect on cargo binding, suggesting a less critical role in the autoinhibition mechanism. The SNX17$_{CT-18}$ peptide, covering the C-terminal 18 amino acids with the NxxF motif, binds weakly to SNX17, confirming the self-interaction, with the N459A + F462A mutation significantly enhancing binding affinity, while W321A and V380D mutations prevent this interaction. These results indicate that the last 18 residues of SNX17 occupy the cargo binding site, thus validating the AF2-based structural model and supporting the proposed autoinhibitory conformation.

To assess if SNX17 autoinhibition affects Retriever recruitment, we conducted pull-down assays with SNX17 mutants under both cargo-present and cargo-absent conditions (Fig. 4F). Mutations W321A, V380D, and H457A notably increased Retriever binding in the absence of cargo, indicating that disrupting autoinhibition exposes the CT region, bypassing the need for cargo-induced release. In contrast, the N459A + F462A mutation did not enhance Retriever binding, as these residues are involved in the interaction with the VPS26C-VPS35L interface according to the AF2-multimer models (Fig. 3B). These results support the proposed autoinhibition state of SNX17 for Retriever association, which is unlocked through cargo binding (Fig. 4D).

SNX31, a protein closely related to SNX17, contains a PX domain, a FERM domain, and a C-terminal unstructured region. Similar to SNX17, SNX31 associates with Retriever via its terminal leucine (McNally et al, 2017). The C-terminal tail of SNX31 might also interact with its cargo-binding pocket, adopting an autoinhibited conformation. It will be important to confirm whether cargo binding also plays a regulatory role in Retriever recruitment by SNX31.

## SNX17 interaction with vesicles triggers Retriever recruitment

The mechanism of Retriever recruitment to endosomal membranes remains unclear, particularly whether Retriever binds directly to membranes or relies on SNX17 for cargo-dependent recruitment. To address this, we performed membrane-binding studies using artificial vesicles containing phosphatidylinositol 3-phosphate (PI3P), which binds the PX domain of SNX17 (Fig. 5A; Appendix Fig. S12). Using giant unilamellar vesicles (GUVs), which are easily imaged by confocal

fluorescence microscopy, we observed that GFP-SNX17 bound over the entire GUV surface, while mKate2-Retriever was not detected bound to the GUV membranes on its own. However, in the presence of GFP-SNX17, mKate2-Retriever was successfully recruited to the GUV membrane. Notably, adding the cargo His$_{10}$-L2$_{FBR}$ did not significantly increase Retriever binding to the GUVs. These results suggest that membrane association induces a conformational change in the SNX17 CT region that enables Retriever binding (Fig. 5B). To support this mechanism, we evaluated previously studied SNX17 mutants of the Retriever-binding motif (SNX17$_{D467X}$ and SNX17$_{L470G}$), which failed to recruit Retriever to membranes, indicating that the GUV-based assay recreates the Retriever-SNX17 binding mode observed in solution, mediated through the C-terminal end of SNX17. Additionally, autoinhibition-related mutants (SNX17$_{W321A}$, SNX17$_{V380D}$, and SNX17$_{H457A}$) successfully recruited Retriever, while the SNX17$_{N459A}$$_{+F462A}$ mutant failed, as these mutations compromise residues essential for Retriever binding (Fig. EV4A).

To further characterize and quantify Retriever recruitment to membranes, we performed liposome co-sedimentation assays with PI3P-enriched large unilamellar vesicles (LUVs) (Figs. 5C and EV4B). SNX17 alone bound efficiently to PI3P-containing liposomes (64 ± 17%), while Retriever showed minimal nonspecific binding (8.3 ± 2.8%), similar to the sedimented amount without liposomes (6.0 ± 5.4%), likely due to protein aggregation during the assay. However, Retriever sedimentation with LUVs increased significantly in the presence of SNX17 (22 ± 7.0%), consistent with the GUV results. The addition of cargos LRP1 or L2 did not yield a statistically significant increase in co-sedimentation for either SNX17 or Retriever. However, a significant increase in Retriever recruitment to liposomes was observed when SNX17 was preincubated with the cargo LRP1, (Fig. 5C) but not with L2 (Fig. EV4B), likely due to variability between replicates. These findings suggest that, within a membrane context, as observed in solution, the presence of cargo can further enhance Retriever recruitment. Our results indicate that SNX17 binding to PI3P-containing membranes facilitates Retriever binding to SNX17 in the absence of cargo, with cargo potentially providing additional enhancement.

Recent studies highlight the role of PIKfyve, a lipid kinase that uses the pool of PI3P to generate PI(3,5)P$_2$ (phosphatidylinositol 3,5-bisphosphate), in the Retriever-dependent recycling of integrins on endosomes (Giridharan et al, 2022). PI(3,5)P$_2$ production is proposed to facilitate Retriever and CCC complex recruitment to

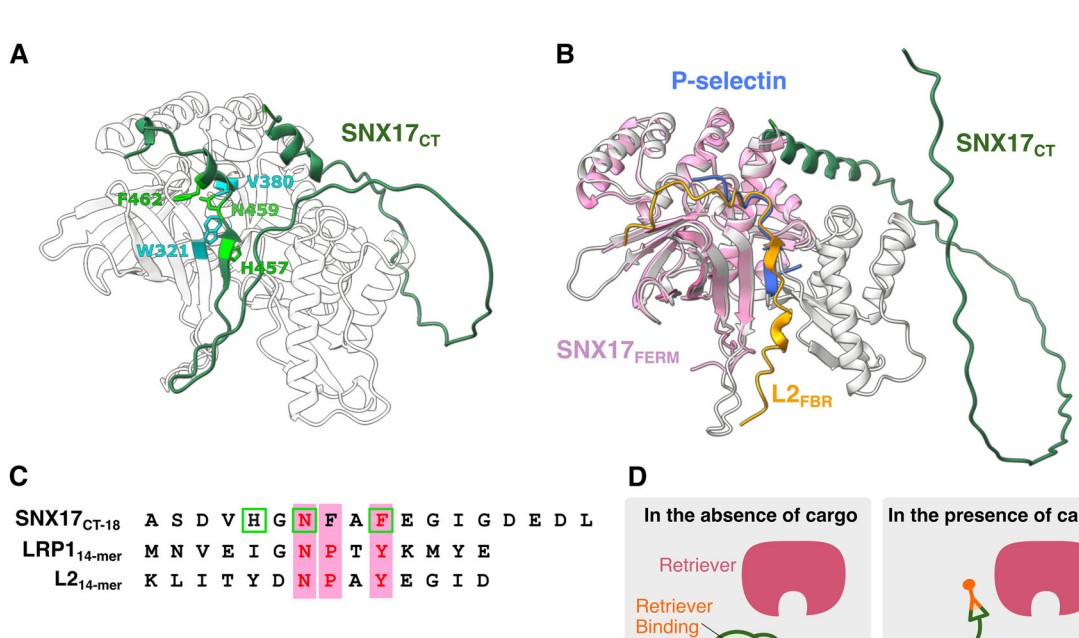

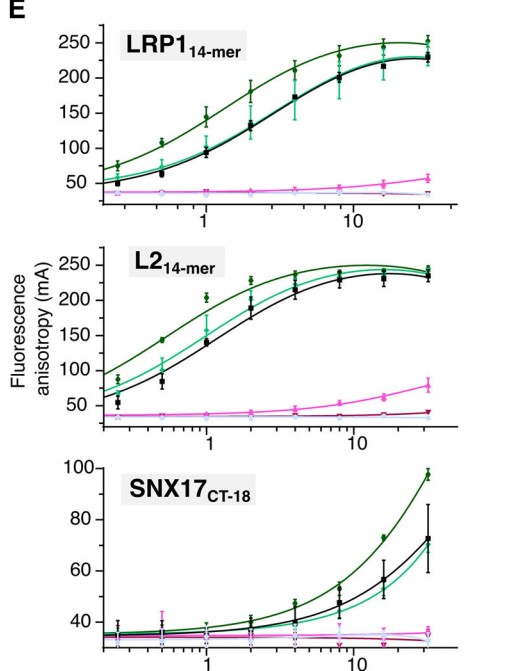

**C**

| SNX17<sub>CT-18</sub> | A S D V H G N F A F E G I G D E D L |
| LRP1<sub>14-mer</sub> | M N V E I G N P T Y K M Y E |
| L2<sub>14-mer</sub> | K L I T Y D N P A Y E G I D |

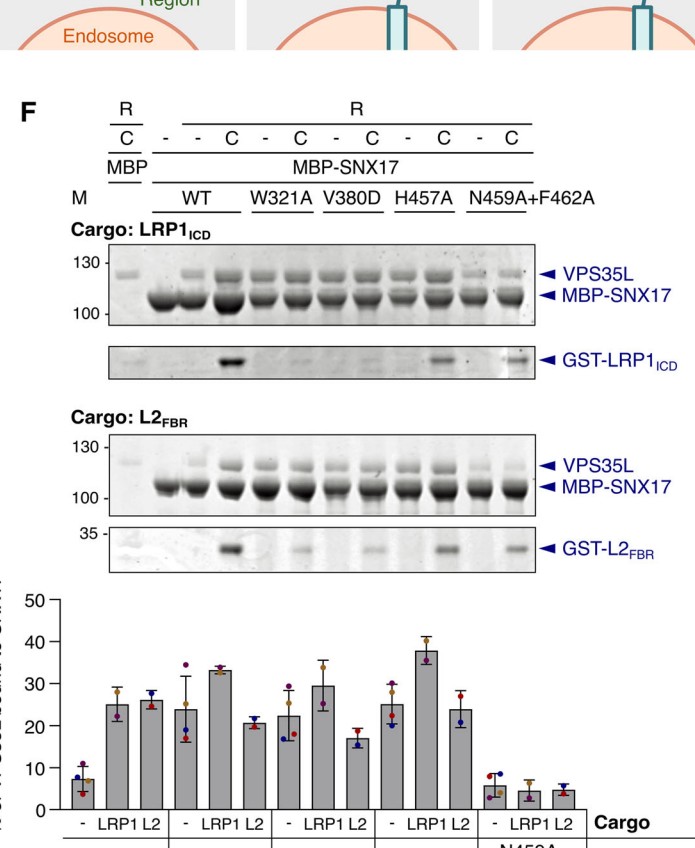

**D**

In the absence of cargo | In the presence of cargo | Retriever recruitment

**E**

Fluorescence anisotropy (mA)

log [MBP/MBP-SNX17](μM)

| MBP-SNX17 | $K_D$ (μM) | | |
|---|---|---|---|
| | LRP1<sub>14-mer</sub> | L2<sub>14-mer</sub> | SNX17<sub>CT-18</sub> |
| WT | 3.0 ± 0.2 | 1.1 ± 0.1 | LB |
| W321A | LB | 76 ± 7 | NB |
| V380D | NB | NB | NB |
| H457A | 3.8 ± 2.2 | 1.1 ± 0.3 | LB |
| N459A+F462A | 1.5 ± 0.5 | 0.6 ± 0.1 | 86 ± 6 |
| MBP | NB | NB | NB |

**F**

**Figure 4. SNX17 autoinhibition mechanism for Retriever binding in the absence of cargo.**

(A) SNX17 residues involved in the autoinhibition mechanism. The CT region of SNX17 (residues 389 to 470) is depicted in dark green. Key amino acids are highlighted with sticks; W321 and V390 residues belong to the cargo-binding pocket (cyan), whereas H457, N459, and F462 residues are part of the cargo-mimicking region (light green). (B) Alignment of the SNX17$_{FERM}$:P-selectin crystal structure (PDB ID: 4GXB) with the SNX17-L2$_{FBR}$ AF2 model. SNX17 (gray) aligns with SNX17$_{FERM}$ (pink), and the L2$_{FBR}$ peptide (orange) occupies the same position in the cargo-binding pocket as the P-selectin peptide (blue). In the presence of L2 cargo, the CT region of SNX17 (green) is positioned distantly from the cargo-binding pocket, thus making the CT region accessible for potential interactions with other proteins. (C) Sequence alignment of SNX17$_{CT-18}$ with the FERM binding motifs of LRP1 and L2. Residues targeted in mutagenesis studies are marked with green squares. (D) Diagram illustrating the potential autoinhibition mechanism of SNX17. In the absence of cargo, SNX17 exhibits minimal affinity for Retriever, because the cargo-mimicking region (depicted as a triangle) is bound to the cargo-binding pocket rendering the Retriever-binding region poorly accessible (left scene). Cargo binding displaces the inhibitory cargo-mimicking region from the pocket, freeing the Retriever-binding region (middle scene) and facilitating its association with Retriever (right scene). The Retriever-binding region and the cargo-mimicking region partially overlap. (E) Fluorescence anisotropy binding curves upon titration of indicated SNX17 mutants to 5-FAM-labeled LRP1$_{14-mer}$, L2$_{14-mer}$, or SNX17$_{CT-18}$ peptide to validate the autoinhibition mechanism. Data points represent the mean ± SD from three biological replicates, with MBP-SNX17 and its mutants obtained from three independent protein purifications. MBP protein was used as a negative control. $K_D$ values ± SD were determined from individual binding curves. (F) The purified Retriever complex was incubated with the indicated MBP-SNX17 mutants in the presence or absence of GST-LRP1$_{ICD}$ or GST-L2$_{FBR}$ in MBP pull-down assays. Non-fused MBP protein was used as a negative control. Quantification of the Coomassie-stained SDS-PAGE gel was carried out in ImageJ, measuring VPS35L as a representative band of the Retriever complex. The level of Retriever binding to MBP-SNX17 was quantified as described in Fig. 2D. Values represent mean ± SD ($n = 2$ biological replicates, with MBP-SNX17 and its mutants obtained from two independent protein purifications. R retriever, C cargo, WT wild-type, NB no detectable binding, LB low binding (poor fit quality). Source data are available online for this figure.

endosomes. To test whether PI(3,5)P$_2$ or its derivative PI5P contributes to Retriever binding, we conducted co-sedimentation assays, either excluding PIPs (phosphatidylinositol phosphates) (Fig. EV4C) or replacing PI3P with PI5P (Fig. EV4D) or PI(3,5)P$_2$ (Fig. EV4E). Liposomes lacking PIPs or containing PI5P or PI(3,5)P$_2$ did not recruit SNX17 or Retriever, suggesting that PI(3,5)P$_2$ and PI5P have no direct effect on Retriever recruitment. Whether other lipid compositions influence the recruitment of the Retriever complex to the membrane requires further investigation.

In our defined, endosome-mimicking system, only PI3P-enriched membranes effectively recruited SNX17. This suggests that the interaction with PI3P induces a conformational change in SNX17, similar to that triggered by cargo binding. This conformational change disengages the CT region from the cargo-binding pocket, exposing the Retriever-binding region at the C-terminus. Additionally, the induced conformational change in the CT region of SNX17 may contribute to enhanced cargo binding (Fig. 5B).

Previous studies have suggested that Retriever recruitment to endosomes is mediated by its association with the CCC complex, which itself is recruited to endosomes by the WASH complex. McNally et al reported that depletion of SNX17 does not affect the recruitment of Retriever to membranes (McNally et al, 2017). Similarly, Giridharan et al found that the presence of SNX17 and WASH complex at endosomes is not sufficient to recruit either the Retriever or CCC complexes (Giridharan et al, 2022). However, our in vitro studies provide evidence that SNX17 plays a crucial role in Retriever recruitment to membranes. Our data indicate that Retriever lacks intrinsic membrane-binding affinity for liposomes, suggesting that SNX17 acts as an anchor for Retriever's engagement with membranes. There is good evidence that the CCC complex enhances Retriever recruitment to endosomes, but it is not indispensable for this process. Singla et al observed increased cytosolic staining of VPS35L following COMMD3 or CCDC93 deficiency, without completely preventing endosomal recruitment of VPS35L (Singla et al, 2019). Boesch et al also found that VPS35L variants, which disrupt CCC interaction, still retain endosomal localization in cellular studies (Boesch et al, 2023). Based on these findings, we suggest that the interaction between SNX17 and

Retriever might be sufficient for the recruitment of the Commander complex to endosomes and that CCC and WASH complexes act as reinforcement.

Concurrent with our work, two independent groups have provided complementary structural and functional insights into the Retriever-SNX17 complex (Butkovič et al, 2024; Singla et al, 2024). Our AF2-predicted Retriever-SNX17 model aligns with the recently solved cryo-EM structure, which shows the SNX17 C-terminal tail engaging the VPS26C-VPS35L interface (Singla et al, 2024). Both studies demonstrated that disrupting this interface impairs the cellular recycling of SNX17-dependent cargos, emphasizing the biological relevance of our findings. However, Singla et al reported high affinity between SNX17 and Retriever without cargo, a discrepancy likely due to their use of non-physiological salt concentrations (50 mM NaCl) in their assays, as we experimentally confirmed (Fig. EV2C). Our study describes an autoinhibited conformation and cargo-dependent activation mechanism for SNX17 that closely aligns with the findings of Butkovič et al. Furthermore, we provide key additional insights, identifying SNX17 membrane interaction as an additional trigger for Retriever recruitment.

In summary, this study provides a comprehensive model for the recruitment of the Retriever complex to membranes. We suggest that the autoinhibitory conformation of SNX17 prevents the assembly of the entire recycling machinery when SNX17 is in the cytoplasm or loosely associated with nonspecific membranes. This mechanism could act as a spatiotemporal control, ensuring that SNX17 is exclusively activated at endosomal membranes for Retriever recruitment. SNX17 holds the recycling system in an "off" state, which can be switched to an "on" state by two different mechanisms: either through selective engagement with a cargo protein or by a specific association with membranes containing PI3P (Fig. EV5). We propose that integrating both Retriever recruitment mechanisms might be essential for simultaneous membrane coating and cargo selection. This model illustrates the initial step of cargo incorporation into a presently unidentified membrane transport carrier of the SNX17-Retriever-CCC pathway for transporting the cargo from the endosome to the plasma membrane. Further research is required to decipher the roles of other components of the Commander complex, such as the CCC

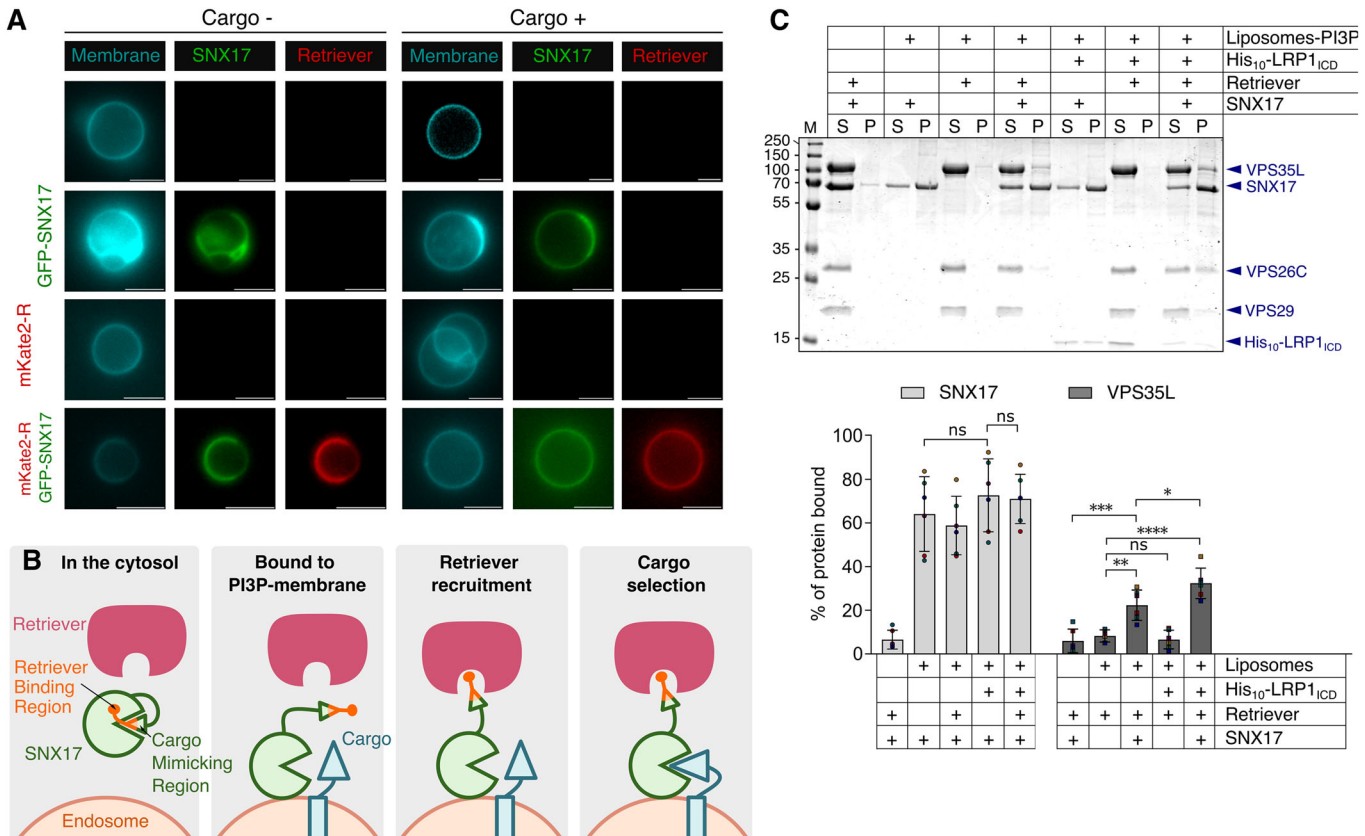

**Figure 5. SNX17 interaction with vesicles triggers Retriever recruitment.**

(A) Confocal fluorescence imaging of GUVs incubated with Retriever-mKate2 (red), GFP-SNX17 (green), or both in the presence or absence of the cargo His$_{10}$-L2$_{FBR}$. GUV membranes were stained with Marina Blue DHPE lipid dye (cyan). Scale bar, 5 µm. (B) Diagram of the potential mechanism of Retriever recruitment via SNX17 in membranes. In the absence of endosomal membrane and cargo, the CT region of SNX17 interacts with itself and masks the Retriever-binding region. Under this condition, SNX17 has a negligible affinity for Retriever. After membrane attachment, the CT region of SNX17 becomes exposed, allowing both Retriever recruitment and cargo binding to facilitate the assembly of the recycling machinery. (C) A co-sedimentation assay was performed by incubating PI3P-containing liposomes with His-Sumo3-SNX17, Retriever, and His$_{10}$-LRP1$_{ICD}$ (at a 2:2:4 µM ratio). Samples were loaded onto an SDS-PAGE gel to separate the soluble (supernatant, S) and co-sedimented (pellet, P) fractions (upper panel). After Coomassie staining, densitometry-based quantification of the individual bands was performed. The binding of SNX17 and Retriever to liposomes was quantified as the percentage of total protein bound to the pellet under each condition, with VPS35L serving as a representative band of the Retriever complex. The values in the graph (bottom panel) represent the mean ± SD of six biological replicates, derived from three independent liposome preparations and two separate protein purifications of Retriever and His-Sumo3-SNX17. Statistical significance was tested using one-way ANOVA followed by Tukey's test for multiple comparisons. *$p = 0.03$, **$p = 0.002$, ***$p = 0.0002$, ****$p = 0.0000006$, ns not significant. Source data are available online for this figure.

complex and DENND10 in this initial step of endosomal membrane association for recycling transmembrane cargos from endosomes to the cell surface.

# Methods

## Reagents and tools table

| Reagent/resource | Reference or source | Identifier or catalog number |
|---|---|---|
| **Experimental models** | | |
| *Escherichia coli* Top10 | Thermo Fisher | Cat# C404003 |

| Reagent/resource | Reference or source | Identifier or catalog number |
|---|---|---|
| *Escherichia coli* BL21 (DE3) | Thermo Fisher | Cat# C600003 |
| *Escherichia coli* DH10EMBacY | Geneva Biotech | |
| Sf9 | Thermo Fisher | 12659017 |
| High Five | Thermo Fisher | B85502 |
| HEK293T | ATCC | CRL-3216 |
| **Recombinant DNA** | | |
| Plasmid constructs | This study | Appendix Table S2 |
| **Oligonucleotides and other sequence-based reagents** | | |
| PCR primers | This study | Appendix Table S3 |

| Reagent/resource | Reference or source | Identifier or catalog number |
|---|---|---|
| **Antibodies** | | |
| Rabbit Polyclonal anti-SNX17 | Sigma | Cat# HPA043867 |
| Rabbit Polyclonal anti-VPS35L | Invitrogen | Cat# PA528553 |
| Mouse Monoclonal anti-Flag M2 | Sigma | Cat# F1804 |
| Mouse Monoclonal anti-beta Actin | Santa Cruz | Cat# sc-47778 |
| Goat IRDye800CW anti-rabbit IgG | LI-COR | Cat# 926-32211 |
| Goat IRDye680RD anti-mouse IgG | LI-COR | Cat# 926-68070 |
| Goat HRP anti-rabbit IgG | Bio-Rad | Cat# 1705046 |
| **Chemicals, enzymes, and other reagents** | | |
| Recombinant proteins | This study | Appendix Table S4 |
| Peptide LRP1$_{14mer}$ (MNVEIGNPTYKMYE) | Genscript | N/A |
| Peptide APP$_{14mer}$ (QQNGYENPTYKFFE) | Genscript | N/A |
| Peptide ITGB1$_{P-14mer}$ (KWDTGENPIYKSAV) | Genscript | N/A |
| Peptide ITGB1$_{D-13mer}$ (AVTTVVNPKYEGK) | Genscript | N/A |
| Peptide L2$_{14mer}$ (KLITYDNPAYEGID) | Genscript | N/A |
| Peptide SNX17$_{CT-18}$ (MNVEIGNPTYKMYE) | Genscript | N/A |
| HyClone SFX-Insect cell culture medium | Cytiva | Cat# HYCLSH30278 |
| FuGENE HD Transfection Reagent | Promega | Cat# E2311 |
| FBS | Gibco | Cat# A5256701 |
| X-Gal | VWR | Cat# 437132J |
| IPTG | Fisher | Cat# BP175510 |
| PMSF | VWR | Cat# 0754-5G |
| Benzamidine | Thermo Scientific | Cat# 401790250 |
| DNAse | PanReac | Cat# A3778 |
| Lysozyme | Sigma | Cat# 62971 |
| DTT | Thermo Scientific | Cat# R0862 |
| TCEP | GoldBio | Cat# TCEP |
| Trehalose | Sigma | Cat# T9449 |
| Raffinose pentahydrate | Thermo Scientific | Cat# A18313 |
| Triton X-100 | Thermo Scientific | Cat# A16046.AP |
| Tween-20 | Panreac | Cat# A4974 |
| PureCube Glutathione Agarose | Cube Biotech | Cat# 32105 |
| PureCube Ni-INDIGO Agarose | Cube Biotech | Cat# 75105 |
| Streptactin-XT-4Flow Resin | IBA Lifesciences | Cat# 2-5030-010 |
| Amylose Resin High Flow | New England Biolabs | Cat# E8022S |

| Reagent/resource | Reference or source | Identifier or catalog number |
|---|---|---|
| HitrapQ | Cytiva | Cat# 17115401 |
| Superdex75 10/300 GL | Cytiva | Cat# 17517401 |
| Superdex200 Increase 10/300 GL | Cytiva | Cat# 28990944 |
| HiLoad 16/600 Superdex200 pg | Cytiva | Cat# 28989335 |
| Imidazole | Sigma | Cat# 56750 |
| L-Glutathione reduced | GoldBio | Cat# G-155 |
| Biotin | IBA Lifesciences | Cat# 21016005 |
| Marina Blue™ DHPE lipid dye | Invitrogen | Cat# M12652 |
| 1,2-dioleoyl-sn-glycerol-3-phosphocholine (DOPC) | Avanti Polar Lipids | Cat# 850375 |
| 1,2-dioleoyl-sn-glycerol-3-phosphoethanolamine (DOPE) | Avanti Polar Lipids | Cat# 850725 |
| 1,2-dioleoyl-sn-glycerol-3-phosphoserine (DOPS) | Avanti Polar Lipids | Cat# 840035 |
| 1,2-dioleoyl-sn-glycero-3-phospho-(1'-myo-inositol-3'- phosphate) (18:1 PI3P) | Avanti Polar Lipids | Cat# 850150 |
| 1,2-dioleoyl-sn-glycero-3-phosphoethanolamine-N-(lissamine rhodamine B sulfonyl) (18:1 Liss Rhod PE) | Avanti Polar Lipids | Cat# 810150 |
| 40 µm plain silica microspheres | Corpuscular Inc | Cat# C-SIO-40.0 |
| NuPAGE 4–12% Bis-Tris SDS-PAGE gel | Invitrogen | Cat# WG1403BOX |
| PageRuler™ Plus Prestained Protein Ladder, 10 to 250 kDa | Thermo Scientific | Cat# 26619 |
| µ-Slide 18-well uncoated chambered coverslip | ibidi | Cat# 81811 |
| PVDF Immobilon-FL membrane | Millipore | Cat# IPFL00005 |
| Whatman gel blotting paper, Grade GB003 | Cytiva | Cat# 10427812 |
| Intercept (PBS) Blocking buffer | LI-COR | Cat# 927-70001 |
| Gel ANTI-FLAG M2 Affinity | Sigma | Cat# A2220 |
| NZY Advanced ECL | NZYtech | Cat# MB40201 |
| **Software** | | |
| ATSAS 4.0 | (Manalastas-Cantos et al, 2021) | |
| BioXtas RAW 2.3.0 | (Hopkins et al, 2017) | |
| Prism 9 | GraphPad | |
| ImageJ /Fiji 2.14.0 | https://imagej.net/software/fiji/ | |
| Empiria Studio Software | LI-COR | |
| USCF ChimeraX 1.18 | (Pettersen et al, 2021) | |

| Reagent/resource | Reference or source | Identifier or catalog number |
|---|---|---|
| Colabfold | (Mirdita et al, 2022) | |
| Consurf | (Ashkenazy et al, 2016) | |
| PROMALS3D | (Pei et al, 2008) | |
| ESPript 3 | (Robert and Gouet, 2014) | |
| **Other** | | |
| Tecan Spark 10 M Plate Reader | Tecan | |
| Odyssey CLx Imager | LI-COR | |
| ECLIPSE Ti2 inverted microscope | NIKON | |
| JASCO J-810 CD spectropolarimeter | JASCO | |

## Strains and plasmids

The *Escherichia coli* strains Top10 and BL21(DE3) were used for cloning and protein expression, respectively. These bacteria were routinely grown at 37 °C in Luria broth (LB) liquid medium with shaking or on LB-agar plates. Insect cells Sf9 and High Five were grown in HyClone SFX-Insect cell culture medium (Cytiva).

For molecular cloning, the Gibson Isothermal DNA Assembly method was used. The nucleotide sequences of VPS26C, VPS35L, and LRP1 were optimized for bacterial expression by the Invitrogen GeneArt Synthesis Service, and the ITGB1 sequence by the IDT Codon Optimization Tool. Site-directed mutagenesis was also performed with the Gibson Isothermal DNA Assembly method using mutagenic primers. A set of cloning plasmids, designated as pIA ("Isothermal Assembly"), were constructed in this study for the efficient cloning of a single PCR product into various vectors using Gibson assembly. These vectors facilitate the expression of the desired sequence with either a His-, GST-, HisMBP-, Strep-, or TwinStrep- tag, along with the TEV (Tobacco Etch Virus), SenP2 (Sentrin-specific Protease 2), or HRV-3C (Human Rhinovirus 3C protease) protease recognition site at the N-terminus. Plasmids used in this study are listed in Appendix Table S2, the sequences of the oligonucleotides and template DNA used for construct generation are summarized in Appendix Table S3, the sequences of the resulting recombinant proteins purified in this study are listed in Appendix Table S4, and the oligonucleotides used for RT-qPCR are specified in Appendix Table S6.

## Generation of recombinant baculovirus

The Retriever subunits were cloned into the pLIB vector and then combined into the pBIG1a plasmid using the biGBac method (Weissmann et al, 2016). The pBIG1a recombinant vectors were introduced by heat shock into DH10EMBacY competent cells to generate recombinant baculoviral genomes via Tn7 transposition. After plating the transformants on agar plates containing 100 µg/ml 5-bromo-4-chloro-3-indolyl-β-D-galactopyranoside (X-gal), 0.1 mM iso-propyl β-D-1-thiogalactopyranoside (IPTG), 50 µg/ml kanamycin, 10 µg/ml tetracycline, and 10 µg/ml gentamycin, white colonies were selected. Bacmid DNA was extracted by lysing the cells with the

GeneJet Plasmid Miniprep Kit solutions, followed by isopropanol precipitation of the supernatant, and washing of the pellet with 70% ethanol. Sf9 cells were seeded at $4 \times 10^5$ cells/well in a six-well plate. Bacmid DNA was transfected into the cells using FuGENE HD Transfection Reagent (Promega) according to the manufacturer's protocol and incubated at 27 °C for at least 72 h. Transfection efficiency was monitored by observing the fluorescence of the YFP protein through fluorescence microscopy. After centrifugation of the cells at $700 \times g$ for 5 min at 4 °C, 2% of fetal bovine serum (FBS) (Gibco, Thermo Fisher Scientific) was added to the supernatant and was then used to infect a 25 ml suspension culture of Sf9 cells at $1 \times 10^6$ cells/ml. At 72 h post-infection, the P1 generation of the virus was harvested by collecting the supernatant again, and 2% FBS was added. For further virus amplification, 2 ml of P1 was used to infect a 100 ml culture of Sf9 cells at $1 \times 10^6$ cells/ml. The supernatant, termed P2, was harvested after 72 h, 2% FBS was added, filtered with 0.2-µm-filters (Whatman GE Healthcare Life Sciences), and stored at 4 °C in the dark.

## Protein purification

For protein overexpression in bacteria, the *Escherichia coli* BL21(DE3) strain was used. Cells were grown in LB medium at 37 °C to an optical density at 600 nm of 0.7–0.9. After cooling for 30 min, protein expression was induced with 0.5 mM IPTG at 18 °C for 16 h. Cells were harvested by centrifugation, and the cell pellet was resuspended in buffer A (50 mM Tris-HCl pH 8.0, 300 mM NaCl, 1 mM DTT) supplemented with 0.5 mM phenylmethylsulfonyl fluoride (PMSF), 5 mM benzamidine, 25 µg/ml DNAse and 1 mg/ml lysozyme. For the purification of proteins containing a His-tag, 20 mM imidazole was also included. After 30 min incubation, the bacteria were disrupted by sonication in an ice bath and cleared by centrifugation at $15,000 \times g$ for 45 min.

For baculovirus-insect cell expression, 250 ml suspension cultures of High Five insect cells at $1 \times 10^6$ cells/ml were infected with P2 baculovirus solution containing the Retriever constructs. At 72- or 96-h post-infection, when YFP fluorescence reached a plateau, cells were harvested by centrifugation at $1000 \times g$ for 5 min at 4 °C. Insect cells were lysed using probe sonication in the same lysis buffer as in bacteria, without the addition of lysozyme. The lysate was ultracentrifuged at $163,000 \times g$ for 45 min.

The lysis supernatant was incubated with PureCube Glutathione agarose (Cube Biotech), PureCube Ni-INDIGO agarose (Cube Biotech), or Streptactin-XT-4Flow beads (IBA Lifesciences GmbH) if the protein of interest had a GST-tag, His-tag, or Strep-tag, respectively. This was followed by extensive washing with buffer A in a gravity column. The protein linker was proteolytically removed by overnight incubation at 4 °C in the presence of SenP2, HRV-3C, or TEV proteases, depending on the cleavage site of the proteins, in 25 mM Tris-HCl pH 8.0, 300 mM NaCl, and 1 mM DTT. For buffer exchange, dialysis was performed with dialysis membranes with a molecular weight cutoff (MWCO) of 6–8 kDa (Spectrum Laboratories Inc.). In this study, tags were retained on certain proteins to enhance their stability and solubility. For further purification, proteins were subjected to ion exchange chromatography (HitrapQ, Cytiva), employing a NaCl gradient ranging from 100 to 1000 mM NaCl. This step was followed by size exclusion chromatography (Superdex75 10/300, Superdex200 10/300, or Superdex200 16/60; Cytiva) in buffer B (25 mM Hepes pH 8.0, 150–300 mM NaCl, and 1 mM TCEP). These

chromatographic separations were performed on an ÄKTA™ Pure protein purification system. The steps performed for the purification of each protein are detailed in Appendix Table S5.

Successful protein purification was confirmed by denaturing polyacrylamide gel electrophoresis (SDS-PAGE). When required, the protein was concentrated using Amicon Ultra Centrifugal filters (Merck Millipore) and quantified by measuring the absorbance at 280 nm and using the theoretical extinction coefficient. The solubility and purification yield of the SNX17 and Retriever mutants were identical to the wild-type proteins, indicating that the mutations do not significantly disrupt their structure.

## SAXS

All SAXS measurements were performed at the B21 bioSAXS beamline, equipped with an EigerX 4M (Dectris) detector, at Diamond Light Source synchrotron, Oxfordshire (Cowieson et al, 2020). SAXS collection details are listed in Appendix Table S1. A Shodex KW-403 gel-filtration column was coupled to SAXS measurement equilibrated in 50 mM Tris pH 7.5, 200 mM NaCl (150 mM NaCl for the VPS26C sample), and 1 mM TCEP. For VPS26C, a total of 620 frames were recorded with an exposure time of 1.0 s/frame using an X-ray wavelength of $\lambda = 0.954$ Å in flow mode at 25 °C. For the Retriever complex, 600 frames were acquired with an exposure time of 3.0 s/frame at 15 °C. 2D to 1D radial averaging was performed using the dedicated DAWN software. 1D scattering intensities of the SEC-SAXS data were computed as $I(q)$ versus $q$, where $q = (4 \pi^* \sin\theta)/\lambda$ with $2\theta$ defined as the scattering angle and $\lambda$ the X-ray wavelength. Buffer subtraction, quality assessment, and subsequent analysis were done using ATSAS (Manalastas-Cantos et al, 2021) and BioXtas RAW (Hopkins et al, 2017). Calculation of the forward scattering, $I(0)$, and the radius of gyration, $R_g$, was determined through Guinier analysis and the implementation of the inverse Fourier transformation method of GNOM. The $D_{max}$ derived from the pair-distance distribution of GNOM was also obtained. Ab initio modeling approaches were used to reconstruct low-resolution bead models of the SAXS samples. The VPS26C protein and the whole Retriever complex were modeled using GASBOR. For Retriever the GASBOR algorithm was run 15 times independently and the best-scoring model was chosen. For VPS26C the 15 obtained models were averaged using DAMAVER. The averaged model was then filtered via DAMSTART and finally refined with DAMMIN. Envelopes of the ab initio models were generated using the Molmap function of UCSF ChimeraX. AF2 models and cryo-EM structures were superimposed into the GASBOR-generated envelopes for visual analysis. The protein loops absent in the cryo-EM structure of the Retriever complex (PDB ID: 8SYN) were modeled using AF2. To further assess our data, theoretical scattering profiles for the VPS26C and Retriever AF2 models and the Retriever cryo-EM structure were generated and evaluated against the experimental scattering curves using CRYSOL. UCSF ChimeraX was used for all visualization purposes (Pettersen et al, 2021).

## Fluorescence anisotropy assay

The peptides used for binding assays were synthesized with an N-terminal 5-Carboxyfluorescein (FAM) and HPLC purified (≥95%) by GenScript. Lyophilized peptides were resuspended in

100 mM Hepes pH 8.0 at 5 mg/ml and further diluted in the assay buffer (50 mM Hepes pH 7.5, 300 mM NaCl, and 1 mM DTT). Fifty-microliter binding reactions were prepared by serial dilutions of purified protein with final concentrations ranging between 0.25 and 128 μM and a fixed ligand concentration of 0.1 μM. The mixture was incubated at 25 °C for at least 30 min and then transferred to 96 Flat Black plates. Fluorescence anisotropy was measured at 25 °C using Spark 10M Plate Reader (Tecan) with a 485/20 excitation filter and a 535/35 emission filter. The dissociation constants were calculated in GraphPad Prism by nonlinear regression fitting of the experimental data to a one-site total binding model. The final $K_D$ measurement is the mean of at least two independent experiments.

## Pull-down assays

For pull-down assays, proteins of interest at 5 μM were incubated in binding buffer (25 mM Hepes pH 7.5, 300 mM NaCl, 1 mM DTT, and 0.01% Triton X-100) either with GST-tagged or MBP-tagged ligands at 2.5 μM in the presence of glutathione agarose beads (Cube Biotech) or amylose resin beads (NEB), respectively. A volume of 50 μl of the mixture, along with 10 μl of pre-equilibrated beads, was incubated on a rotating wheel for 2 h at 4 °C. Beads were washed three times with 0.5 ml of binding buffer and resuspended in loading buffer. Protein controls and resin samples were loaded on a precast Invitrogen NuPAGE 4–12% Bis-Tris SDS-PAGE gel (Thermo Scientific) or a self-made 15% SDS-PAGE gel (Mini-PROTEAN, Bio-Rad), together with PageRuler Plus Prestained Protein Ladder (Thermo Scientific). After Coomassie blue staining, gels were scanned with the Odyssey CLx imaging system. Each pull-down was performed in duplicate or triplicate. Non-fused GST or non-fused MBP proteins were used as negative controls. The quantification of pull-down gel results was conducted by assessing background-subtracted band intensities using Fiji/ImageJ software (Schindelin et al, 2012). Statistical analyses were performed with GraphPad Prism, with an unpaired Student's $t$-test applied for comparisons involving biological replicates from independently purified protein samples.

## Circular dichroism spectroscopy

Far-UV CD spectra were acquired at 25 °C using a JASCO J-810 CD spectropolarimeter. Proteins were dialyzed overnight at 4 °C against 100 mM sodium phosphate buffer, pH 7.5, and measured at a concentration of 2 μM. Data were collected using a 0.1-cm path-length quartz cuvette, scanning from 200 to 260 nm at 0.5 nm intervals with a scanning speed of 50 nm/min. A total of 50 scans were accumulated to generate the final spectrum, which was baseline-corrected by subtracting the buffer spectrum. Ellipticity values were converted to mean residue ellipticity.

## Isothermal titration calorimetry

Isothermal titration calorimetry (ITC) experiments were carried out on a Nano ITC calorimeter (TA Instruments). VPS26C and SNX17$_{FERM-CT}$ were dialyzed overnight at 4 °C against 25 mM Hepes pH 7.5, 150 mM NaCl, and 0.5 mM TCEP, and degassed for 30 min at 25 °C before titration. The titration experiments were conducted at 25 °C with one initial 1.5 μl injection followed by 19

injections of 2.5 μl with a spacing of 300 s between injections. To determine the heat of dilution similar injections of $SNX17_{FERM-CT}$ in buffer were carried out. The resulting titration data were analyzed using NanoAnalyze software (TA Instruments).

## GUV preparation

For the study of Retriever binding to membranes, a GUV lipid mixture was prepared that contained Marina Blue™ DHPE lipid dye, 1,2-dioleoyl-*sn*-glycerol-3-phosphocholine (DOPC), 1,2-dioleoyl-*sn*-glycerol-3-phosphoethanolamine (DOPE), 1,2-dioleoyl-*sn*-glycerol-3-phosphoserine (DOPS), 1,2-dioleoyl-*sn*-glycero-3-phospho-(1′-myo-inositol-3′-phosphate) (18:1 PI3P), with DOPC:DOPE:DOPS:PI3P:Marina Blue DHPE in a 45:29.3:20:5:0.7 molar ratio. All lipids were purchased from Avanti Polar Lipids (Alabaster, AL, USA), and the final mix concentration was prepared at 1.5 mM. First, multilamellar lipid vesicles (MLVs) were prepared. All lipids were mixed, incubated for 1 h at 37 °C, desiccated under vacuum to remove chloroform, and rehydrated with a previously degassed working buffer (20 mM HEPES pH 7.5, 200 mM NaCl, 1 mM TCEP), and the mixture was incubated for 1 h at 60 °C. Argon gas was added when tubes were opened to avoid oxidative damage from air. From MLVs, GUVs were generated following the method developed by Velasco-Olmo et al (Velasco-Olmo et al, 2019) and using 40 μm plain silica microspheres (Corpuscular Inc.). Briefly, 2-μl-drops of the lipid mixture were placed on a Teflon surface. 1 μl of plain silica microspheres was brought into contact with the lipid drops and fell into the drop without the need for pipetting. The drops were dried under vacuum and mixed with 6 μl of 1 M trehalose using a cut plastic pipette tip. The tip was then introduced into a homemade humidity chamber, which consists of a 1.5 ml Eppendorf tube halfway filled with Milli-Q water and with a hole in its cap, and incubated in this chamber for 10 min at 60 °C.

## GUV assays and imaging

Marina Blue™ labeled GUV membranes were incubated with 2 μM protein samples for 15 min at room temperature in binding buffer (20 mM HEPES pH 7.5, 200 mM NaCl, 1 mM TCEP). The mixture was examined by fluorescence microscopy in 18-well uncoated chambered coverslips (Ibidi), which were previously treated with BSA at 1 mg/ml for 2 h. For image acquisition, an ECLIPSE Ti2 inverted microscope (Nikon) was used, equipped with an APO TIRF 60x/1.49 lens, an LED light source and an sCMOS camera (Hamamatsu Orca-Flash4.0). The fluorescence of Marina Blue™ fluorophore was detected using a Zeiss G 365 excitation filter and a BP 445/50 emission filter; of EGFP, with a Zeiss BP 470/40 excitation filter and a BP 525/50 emission filter; and of mKate2, with a Zeiss BP 546/12 excitation filter and BP 575–640 emission filter. Images were processed with Fiji (Schindelin et al, 2012).

## Liposomes preparation

All lipids were purchased from Avanti Polar Lipids, INC. (Alabaster, AL) and dissolved in a final 20:9 molar ratio of chloroform:methanol. The lipid composition of liposomes consisted of DOPC, DOPE, DOPS, 18:1 PI(3)P, and 1,2-dioleoyl-*sn*-glycero-3-phosphoethanolamine-*N*-(lissamine rhodamine B sulfonyl) (18:1 Liss Rhod PE) in a final molar ratio of 44.7:29.3:20.0:5.0:1.0, respectively. Vesicles were prepared using the same thin-film hydration method described for GUVs, followed by extrusion to get a more homogeneous vesicle population. A mini extruder (Avanti Polar Lipids) and Whatman Nucleopore track-etched membranes with a pore size of 0.4 μm (Cytiva) were used to obtain Large Unilamellar Vesicles (LUVs). The extrusion buffer (25 mM Hepes pH 7.5, 1 mM TCEP) was supplemented with 250 mM raffinose pentahydrate to produce vesicles filled with raffinose. These vesicles can be separated from the aqueous solution in co-sedimentation assays by centrifugation (Julkowska et al, 2013). A 2 mM liposome suspension was diluted three times its volume with the working buffer (25 mM Hepes pH 7.5, 200 mM NaCl, 1 mM TCEP) and further ultracentrifuged at $50,000 \times g$, 22 °C for 15 min. The resulting pellet containing the vesicles was resuspended in one volume of the working buffer. Liposomes were stored in argon at 21 °C for a maximum of 7 days.

## Liposome co-sedimentation assays

For co-sedimentation experiments, the working buffer (25 mM Hepes pH 7.5, 200 mM NaCl, 1 mM TCEP), proteins (Retriever, HisSumo-SNX17, and $His_{10}$-$L2_{FBR}$ or $His_{10}$-$LRP1_{ICD}$ in a 2:2:4 μM ratio), and liposomes (1 mM final concentration) were sequentially mixed into 0.5 ml tubes. Proteins were centrifuged at $21,100 \times g$, 4 °C for 30 min to remove possible aggregates before mixing. The mixtures were incubated at room temperature on a rotating wheel for 1 h. Samples were then centrifuged at $16,000 \times g$, 21 °C for 30 min. The obtained pellet was washed with 500 μl of the working buffer without resuspending before centrifuging under the same conditions. The supernatant was collected and mixed with 5x loading buffer (S samples). 500 μl of the supernatant was discarded, and the pellet was resuspended in 1x loading buffer (P samples). Both S and P samples were loaded onto a 15% SDS-PAGE gel together with PageRuler Plus Prestained Protein Ladder (Thermo Scientific) and run at 150 V for 90 min. Protein bands were visualized by Coomassie staining and subsequent imaging in Odyssey CLx. Quantification of signal intensity was determined using Empiria Studio Software (LI-COR). Co-sedimentation experiments were done in triplicate under the same conditions using independent preparations of liposomes and protein purifications. Statistical analyses were performed with GraphPad Prism, using one-way ANOVA followed by Tukey's post hoc test.

## Mammalian cell culture and flag-immunoprecipitation assay

For ectopic expression of L2-FlagS mutants, plasmid expression vectors were transfected using linear polyethylenimine (molecular weight, 25,000; Polysciences Inc.) into HEK293T cells. Cells were harvested after 48 h. Immunoprecipitation of the overexpressed Flag-tagged L2 proteins was performed to assess the interaction with the endogenous SNX17 and Retriever complex. HEK293T cells were incubated in 750 μl of lysis buffer (25 mM Hepes pH 7.5, 150 mM NaCl, 1 mM EDTA, 0.1% IGEPAL, and 1x Protease inhibitor cocktail) for 30 min at 4 °C with rotation. Supernatants were collected after centrifugation ($16,000 \times g$, 10 min, 4 °C) and incubated with 17 μl of anti-Flag M2-agarose beads (Sigma, A2220), previously equilibrated in washing buffer (25 mM Hepes pH 7.5, 150 mM NaCl, 1 mM EDTA, 0.25% IGEPAL, and 1x Protease inhibitor cocktail), under gentle rotation at 4 °C for 2 h. After centrifugation ($1500 \times g$, 5 min, at 4 °C), the supernatant was discarded, and the beads were washed three times

with 500 μl of washing buffer. The beads were resuspended in 30 μl lysis buffer, and proteins were eluted by the addition of 10 μl of LB-SDS 5x. Samples were boiled at 98 °C for 5 min. 10 μl of both the input and the immunoprecipitation samples were loaded onto a 12% SDS-PAGE gel (Mini-PROTEAN, Bio-Rad), together with PageRuler Plus Prestained Protein Ladder (Thermo Scientific). The SDS-PAGE was run at 120 V, and proteins were then transferred to a polyvinylidene fluoride (PVDF) membrane at 100 V for 1 h. After blocking for 1 h in Intercept (PBS) Blocking buffer (LI-COR), blots were probed with antibodies against Flag (Sigma, F1804, 1:500), SNX17 (Sigma, HPA043867, 1:1000), VPS35L (Invitrogen, PA528553, 1:1000), and beta-Actin (Santa Cruz, sc-47778, 1:1000) overnight at 4 °C with the addition of 0.02% Tween-20. Membranes were subsequently washed four times with PBS-T (0.02% Tween-20), incubated with the corresponding secondary antibodies (IRDye680RD anti-mouse IgG, LI-COR, 1:5000; IRDye800CW anti-rabbit IgG, LI-COR, 1:10,000; HRP anti-rabbit IgG, Bio-Rad, 1:10,000), and washed four times again prior to visualization. Uncropped and unprocessed scans of all of the blots are provided in the Source Data Appendix File.

## mRNA Gene expression analysis

Total RNA was isolated using the RNeasy Kit (Qiagen). Complementary DNA (cDNA) was synthesized using the iScript Reverse Transcriptase (Bio-Rad). RT-qPCR was performed using the SYBR Select Master Mix (Applied Biosystems) on a OneStep-Plus detection system (Applied Biosystems). The oligonucleotides used for amplification are listed in Appendix Table S6.

## Modeling with AlphaFold2 multimer

The AlphaFold2 multimer algorithm (Jumper et al, 2021) from the Neurosnap web tool and Colabfold (Mirdita et al, 2022) was used to model the SNX17-cargo (SNX17:L2$_{FBR}$, SNX17:L2$_{14mer}$, SNX17:LRP1$_{14mer}$, SNX17:APP$_{14mer}$, SNX17:ITGB1$_{P-14mer}$, SNX17:ITGB1$_{D-13mer}$, SNX17:SNX17$_{CT-18}$) and SNX17-Retriever (SNX17:VPS26C:VPS35L:VPS29, SNX17$_{CT-18}$:VPS26C:VPS35L$_{110-598}$, SNX17:L2$_{17mer}$:VPS26C:VPS35L$_{110-598}$) structures. Five models were generated without templates, through ten iterative refinement recycles, and relaxed using AMBER. The model with the highest pLDDT score was selected. ChimeraX was used to generate all the figures depicting the predicted protein structures and the corresponding PAE plots (Pettersen et al, 2021).

## Evolutionary conservation analysis

Evolutionary conservation analysis was performed with the Consurf web server (Ashkenazy et al, 2016) using the AF2 models as a query and with default parameters. The results were mapped on the structure. Multiple sequence alignments were generated using the structural based alignment web tool PROMALS3D (Pei et al, 2008) and plotted with ESPript 3 (Robert and Gouet, 2014).

## Data availability

SAXS data and fits were deposited at SASBDB (https://www.sasbdb.org/) (Kikhney et al, 2020) under the accession codes: SASDVR6 (VPS26C) and SASDVQ6 (Retriever).

The source data of this paper are collected in the following database record: biostudies:S-SCDT-10_1038-S44319-024-00340-1.

## Peer review information

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

## Acknowledgements

We are extremely grateful to Victor Campa for his assistance in confocal fluorescence microscopy and the staff of beamline B21 of Diamond Light Source for SAXS measurements. This work was supported by grants from the Agencia Estatal de Investigación of Ministerio de Ciencia e Innovación (MCIN/AEI/10.13039/501100011033) RTI2018-097801-B-I00, PID2021-122611NB-100, and TED2021-129278B-I00 to ML, PID2020-117860GB-I00 to JCA and grant PID2021-127816NB-I00 to DA-J. Moreover, funding by MCIN/AEI/10.13039/501100011033 and by "ESF Investing in your future" to ML (RYC-2016-20342) and AM-G (PRE2019-088459) is acknowledged. DA-J was also funded by Fundación Biofísica Bizkaia, the Basque Excellence Research Centre (BERC) program, and IT1745-22 of the Basque Government. MZ-Z acknowledges support from the Basque Government predoctoral program (PRE_2023_1_0100). IM-G received support from Banco Santander through a Ph.D. contract.

## Author contributions

**Aurora Martín-González**: Conceptualization; Formal analysis; Investigation; Writing—original draft. **Iván Méndez-Guzmán**: Conceptualization; Formal analysis; Investigation; Writing—original draft. **Maialen Zabala-Zearreta**: Conceptualization; Formal analysis; Investigation. **Andrea Quintanilla**: Conceptualization; Formal analysis; Investigation. **Arturo García-López**: Investigation. **Eva Martínez-Lombardía**: Investigation. **David Albesa-Jové**: Conceptualization; Supervision; Funding acquisition. **Juan Carlos Acosta**: Conceptualization; Supervision; Funding acquisition. **María Lucas**: Conceptualization; Formal analysis; Supervision; Funding acquisition; Investigation; Writing—original draft; Project administration; Writing—review and editing.

Source data underlying figure panels in this paper may have individual authorship assigned. Where available, figure panel/source data authorship is listed in the following database record: biostudies:S-SCDT-10_1038-S44319-024-00340-1.

## Disclosure and competing interests statement

The authors declare no competing interests.

# Expanded View Figures

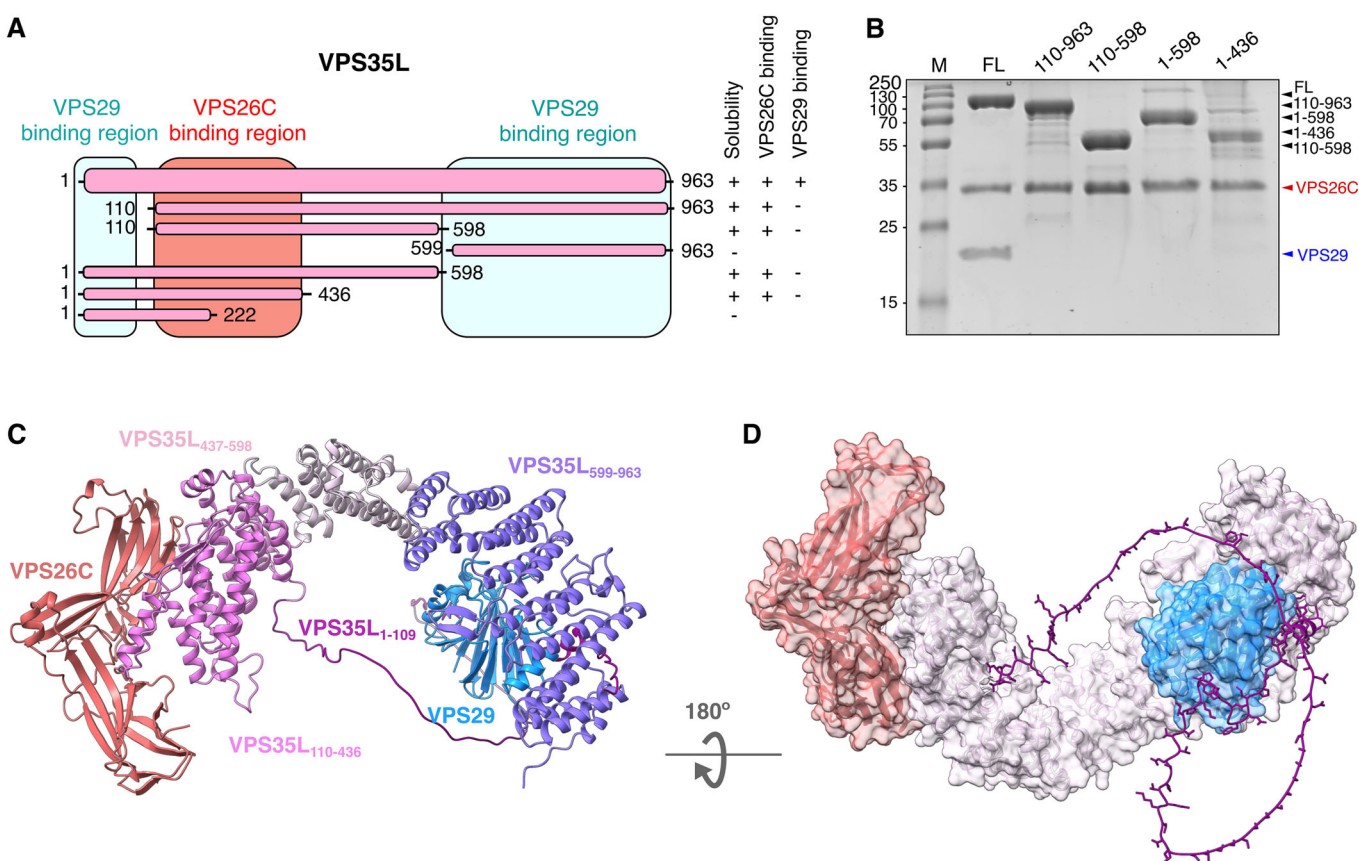

**Figure EV1. Purification and stability assessment of the Retriever complex.**

(A) Summary of the following features of purified Retriever complexes containing VPS26C, VPS29, and different length constructs of VPS35L: solubility, VPS26C binding, and VPS29 binding. The experimentally observed interaction region of VPS35L with VPS26C is highlighted with a red box and with VPS29 with two cyan boxes. (B) Coomassie-stained SDS-PAGE gel of purified Retriever constructs with different VPS35L truncations. (C) AF2 model of the Retriever complex with the experimentally observed regions of interaction highlighted. The interaction region of VPS35L with VPS26C (VPS35L$_{110-436}$) is in pink, the C-terminal interaction region of VPS35L with VPS29 (VPS35L$_{599-963}$) is in violet, and the N-terminal (VPS35L$_{1--109}$) is in dark purple. Model Archive ID: ma-3cag5. (D) Detail of the intramolecular interaction of the amino and carboxy-terminal regions of VPS35L. VPS26C, VPS29, and VPS35L$_{111-963}$ are represented by a ribbon diagram with a transparent surface. VPS35L$_{1-110}$ is displayed in sticks. Source data are available online for this figure.

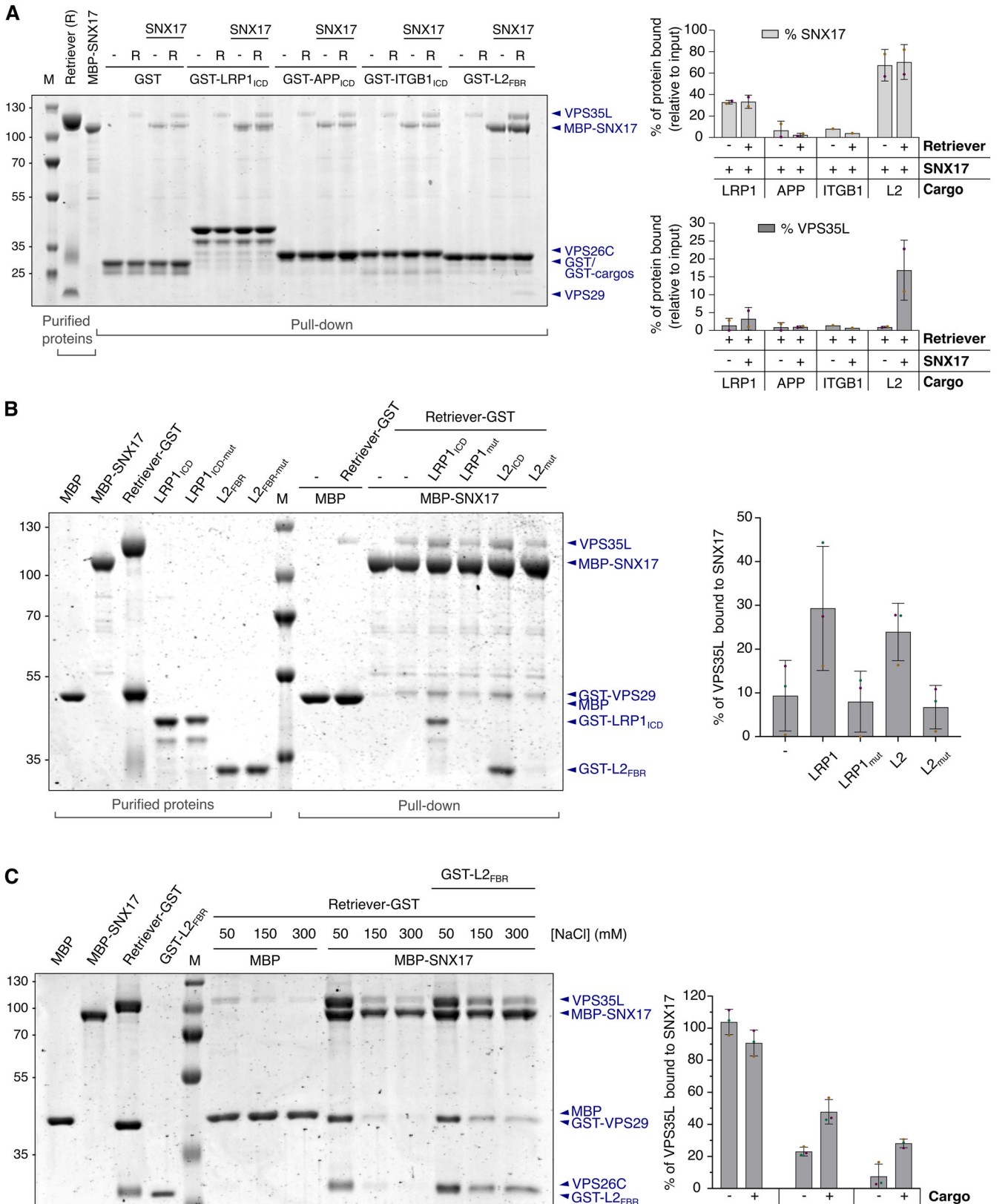

◀ **Figure EV2. Cargo-dependent interaction of SNX17 with Retriever.**

(A) The Retriever complex was incubated with MBP-SNX17 in the presence of GST-LRP1$_{ICD}$, GST-APP$_{ICD}$, GST-ITGB1$_{ICD}$, and GST-L2$_{FBR}$ in GST pull-down assays. Non-fused GST protein was used as a negative control. Purified proteins and pull-down samples were separated by SDS-PAGE and visualized by Coomassie Blue staining (a representative gel shown). The right panel presents the densitometry-based quantification of the amount of SNX17 or Retriever retained in the cargo-GST pull-down assays. VPS35L was used as a representative band of the Retriever complex. The band intensities of SNX17 and VPS35L were normalized to the GST or GST-cargo band intensity. Non-specific binding to GST was subtracted. The percentage of SNX17 or VPS35L binding to GST-cargos was calculated as the ratio of the pull-down protein to the input protein (lanes 2 and 3). Values represent the mean ± SD of two independent experiments. (B) The effect of mutating the conserved NPxY motif to APxA in LRP1 and L2 on the cargo-dependent Retriever-SNX17 interaction. Coomassie-stained SDS-PAGE gel of pull-down assays with MBP-SNX17 and Retriever in the presence of GST-LRP1$_{ICD}$, GST-LRP1$_{ICD-mut}$ (N4470A + Y4473A), GST-L2$_{FBR}$, and GST-L2$_{FBR-mut}$ (N254A + Y257A). Retriever binding to MBP-SNX17 was quantified as described in Fig. 2D. Values represent the mean ± SD of three independent experiments. (C) MBP pull-down assays to examine the impact of salt concentration on the SNX17-Retriever interaction in the presence or absence of cargo. The Coomassie-stained SDS-PAGE gel shown is a representative image of three independent experiments. MBP was included as a control for nonspecific binding. Retriever binding to MBP-SNX17 was quantified as described in Fig. 2D. Error bars represent the standard deviation of three technical replicates. M protein marker, R Retriever. Source data are available online for this figure.

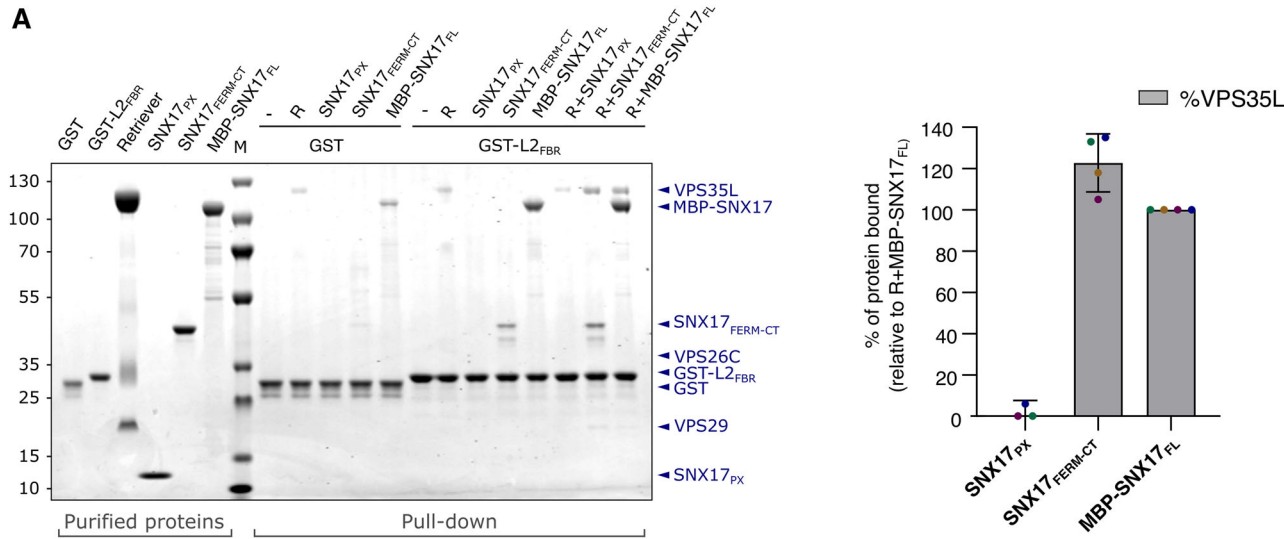

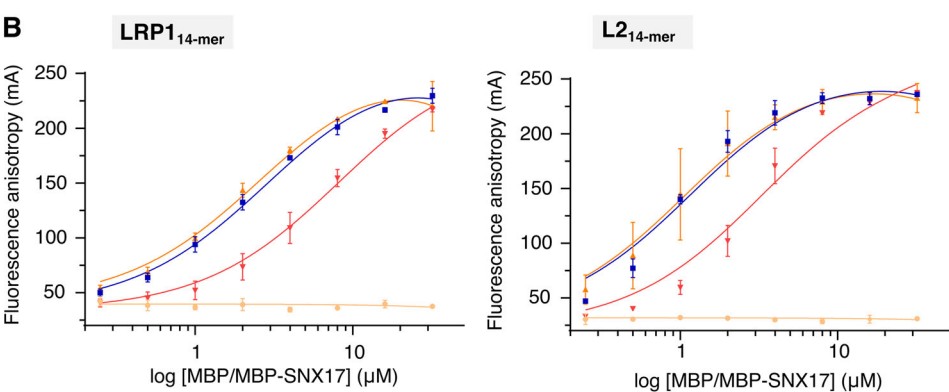

**Figure EV3. Mapping the interaction between SNX17 and the Retriever complex.**

(A) GST pull-down assays to map the region of SNX17 that interacts with Retriever. GST-L2$_{FBR}$ was incubated with the indicated combinations of SNX17$_{PX}$, SNX17$_{FERM-CT}$ and MBP-SNX17$_{FL}$. Non-fused GST protein was used as a negative control. Samples were loaded onto an SDS-PAGE gel and stained with Coomassie Blue. Densitometry-based quantification was carried out with ImageJ, measuring VPS35L as a representative band of the Retriever complex. The band intensities of VPS35L were normalized to the GST or GST-cargo band intensity. Non-specific binding to GST was subtracted. The percentage of VPS35L bound in the presence of MBP-SNX17$_{FL}$ and GST-L2$_{FBR}$ was set to 100%, and the values for the other conditions were calculated relative to this. Values represent mean ± SD based on four technical replicates. M protein marker, R Retriever complex, FL full-length. (B) Effect of SNX17 mutants of the Retriever-binding region on cargo binding affinity. Fluorescence anisotropy binding curves of 5-FAM-labeled LRP1$_{14-mer}$ or L2$_{14-mer}$ peptide titrated with indicated SNX17 mutants. Data points are the mean ± SD of two biological replicates, with MBP-SNX17 and its mutants obtained from two independent protein purifications. The estimated $K_D$ ± SD of each mutant is listed in the right panel. MBP is used as a negative control. NB no detectable binding. Source data are available online for this figure.

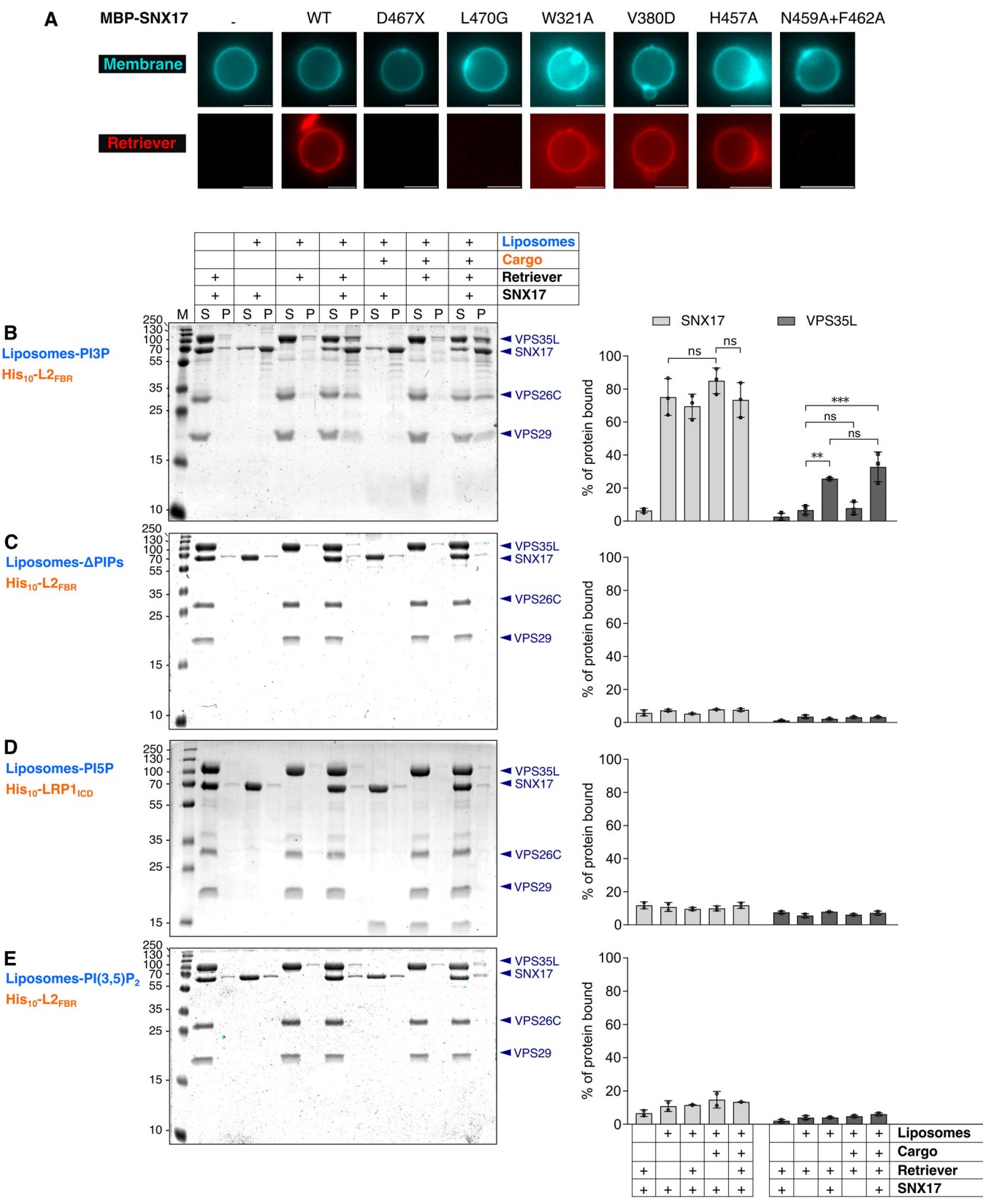

◀ **Figure EV4.   Analysis of the requirements for Retriever recruitment to membranes.**

(A) Fluorescent imaging of GUVs stained with Marina Blue DHPE lipid dye (shown in cyan) to study the interaction of Retriever-mKate2 (red) with SNX17 WT or mutants on membranes. Scale bar: 5 μm. (B–E) Study of Retriever recruitment onto liposome membranes of various compositions in the presence of His-Sumo3-SNX17 and the cargo His$_{10}$-L2$_{FBR}$ or His$_{10}$-LRP1$_{ICD}$. Liposomes lacking phosphatidylinositol (C) or containing PI3P (B), PI5P (D), or PI(3,5)P$_2$ (E) were analyzed. Supernatant (S) and pellet (P) fractions were separated and visualized via SDS-PAGE followed by Coomassie staining (left). The binding of SNX17 and Retriever to liposomes was quantified as the percentage of total protein bound to the pellet in each condition, with VPS35L serving as a representative band of the Retriever complex (right). Bars represent mean ± SD from three (B) or two (C–E) biological replicates, derived from independent liposomes preparations and two separate protein purifications of Retriever and His-Sumo3-SNX17. One-way ANOVA followed by Tukey's test for multiple comparisons was performed for statistical analysis in (B). **$p = 0.004$, ***$p = 0.0003$, ns not significant. Source data are available online for this figure.

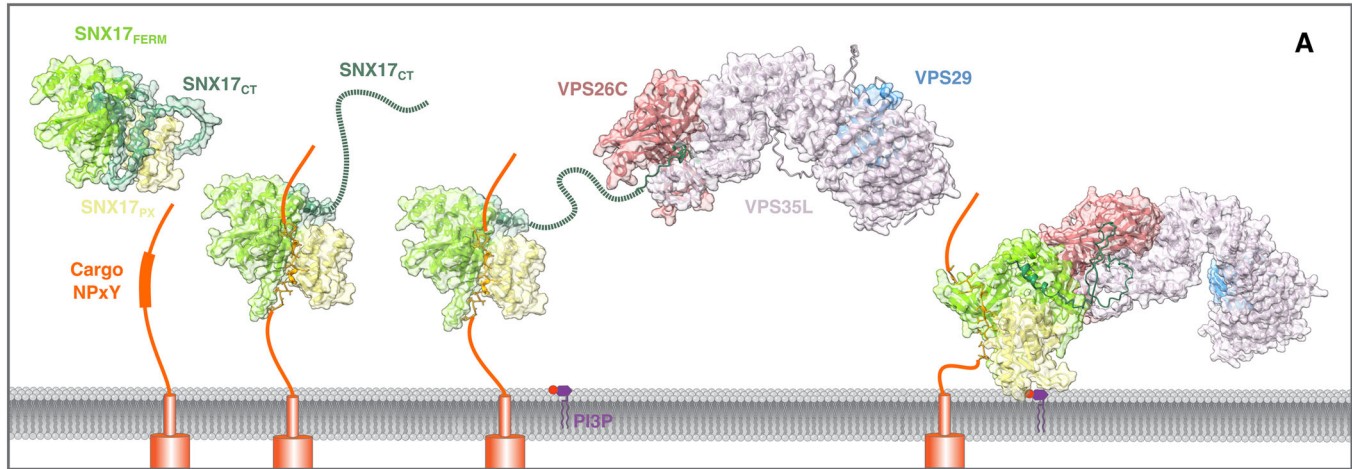

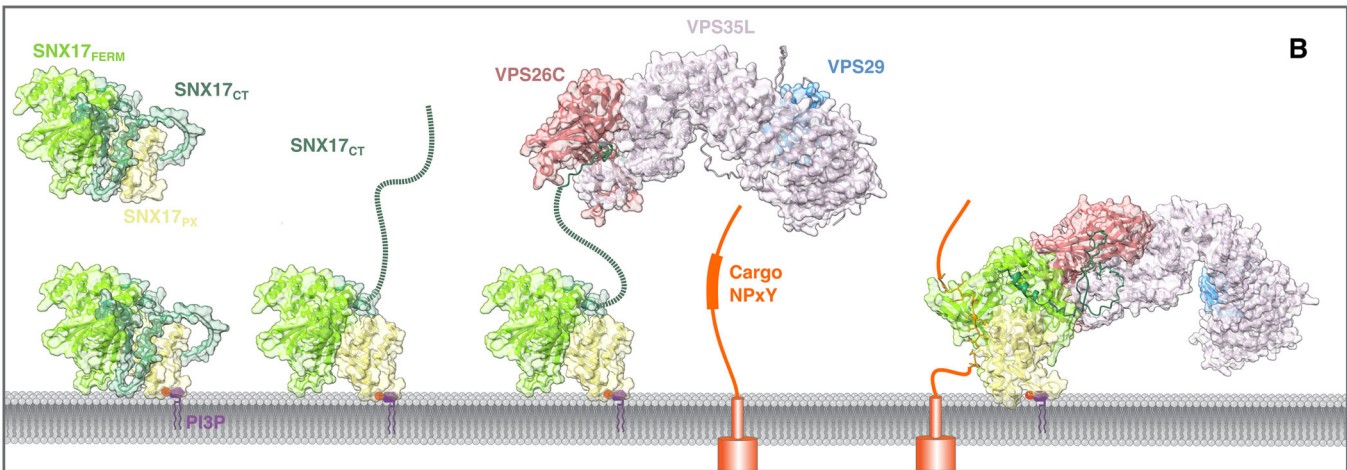

**Figure EV5. Proposed activation mechanisms for the Retriever-SNX17 interaction.**

(A) Cargo-mediated activation: SNX17 encounters its cargo, and this interaction through the FERM domain triggers the release of the SNX17 C-terminal region. With the C-terminal residues exposed, SNX17 binds and recruits Retriever. Subsequently, SNX17 binding to PI3P at the membrane through the PX domain promotes the attachment of the complex to the membrane. (B) Membrane-mediated activation: SNX17 initially binds to PI3P, leading to its attachment to the membrane and subsequent exposure of the Retriever-binding motif. The movement of the C-terminal residues of SNX17 enables Retriever recruitment and cargo binding. The predicted interaction between VPS26C and SNX17, observed in the AF2-multimer model for the complex SNX17:L2$_{17mer}$:VPS26C:VPS35L$_{110-598}$, was used to illustrate the proposed approach of Retriever to the membrane in (A, B).

