## [Peer Review File · EMBO Reports]

Selective cargo and membrane recognition by SNX17 regulates its interaction with Retriever

María Lucas, Aurora Martín-González, Iván Méndez-Guzmán, Maialen Zabala-Zearreta, Andrea Quintanilla, Arturo García-López, Eva Martínez-Lombardía, David Albesa-Jové, and Juan Acosta

Corresponding author(s): María Lucas (maria.lucas@unican.es)

Review Timeline:

Submission Date:	28th Feb 24
Editorial Decision:	25th Mar 24
Revision Received:	30th Aug 24
Editorial Decision:	23rd Sep 24
Revision Received:	9th Nov 24
Accepted:	15th Nov 24

Editor: *Martina Rembold*

Transaction Report:

Dear Dr. Lucas

Thank you for the submission of your research manuscript to our journal. We have now received the full set of referee reports that is copied below.

As you will see, the referees acknowledge that the findings are interesting and that the conclusions are overall supported by the data presented but they also raise a number of concerns and have suggestions how to further strengthen the data. Among other concerns, the referees consider at least a minimal validation of the findings in cell culture essential to test the physiological relevance of the conclusions. The suggested experiments seem feasible and should be performed. The referees also raised the question whether a virally encoded protein would behave the same as an endogenous cargo. Please either follow the suggestion of referee 1 on immobilizing SNX17 or at least discuss potential limitations related to the viral cargo in the manuscript.

Given these constructive comments, we would like to invite you to revise your manuscript with the understanding that the referee concerns (as detailed above and in their reports) must be fully addressed and their suggestions taken on board. Please address all referee concerns in a complete point-by-point response. Acceptance of the manuscript will depend on a positive outcome of a second round of review. It is EMBO Reports policy to allow a single round of revision only and acceptance or rejection of the manuscript will therefore depend on the completeness of your responses included in the next, final version of the manuscript.

We realize that it is difficult to revise to a specific deadline. In the interest of protecting the conceptual advance provided by the work, we recommend a revision within 3 months (June 25). Please discuss the revision progress ahead of this time with the editor if you require more time to complete the revisions.

I am also happy to discuss the revision further via e-mail or a video call, if you wish.

IMPORTANT NOTE:

We perform an initial quality control of all revised manuscripts before re-review. Your manuscript will FAIL this control and the handling will be delayed IN CASE the following APPLIES:

- 1) A data availability section providing access to data deposited in public databases is missing. If you have not deposited any data, please add a sentence to the data availability section that explains that.
- 2) Your manuscript contains statistics and error bars based on $n=2$. Please use scatter blots in these cases. No statistics should be calculated if $n=2$.

- 1) a .docx formatted version of the manuscript text (including legends for main figures, EV figures and tables). Please make sure that the changes are highlighted to be clearly visible.
- 2) individual production quality figure files as .eps, .tif, .jpg (one file per figure). Please download our Figure Preparation Guidelines (figure preparation pdf) from our Author Guidelines pages <https://www.embopress.org/page/journal/14693178/authorguide> for more info on how to prepare your figures.
- 3) a .docx formatted letter INCLUDING the reviewers' reports and your detailed point-by-point responses to their comments. As part of the EMBO Press transparent editorial process, the point-by-point response is part of the Review Process File (RPF), which will be published alongside your paper.
- 4) a complete author checklist, which you can download from our author guidelines

(<<https://www.embopress.org/page/journal/14693178/authorguide>>). Please insert information in the checklist that is also reflected in the manuscript. The completed author checklist will also be part of the RPF.

5) Please note that all corresponding authors are required to supply an ORCID ID for their name upon submission of a revised manuscript (<<https://orcid.org/>>). Please find instructions on how to link your ORCID ID to your account in our manuscript tracking system in our Author guidelines (<<https://www.embopress.org/page/journal/14693178/authorguide#authorshipguidelines>>)

6) We replaced Supplementary Information with Expanded View (EV) Figures and Tables that are collapsible/expandable online. A maximum of 5 EV Figures can be typeset. EV Figures should be cited as "Figure EV1, Figure EV2" etc... in the text and their respective legends should be included in the main text after the legends of regular figures.

7) Please note that a Data Availability section at the end of Materials and Methods is now mandatory. In case you have no data that requires deposition in a public database, please state so instead of refereeing to the database. See also < <https://www.embopress.org/page/journal/14693178/authorguide#dataavailability>>. Please note that the Data Availability Section is restricted to new primary data that are part of this study.

Additional information on source data and instruction on how to label the files are available <<https://www.embopress.org/page/journal/14693178/authorguide#sourcedata>>.

10) Figure legends and data quantification:
The following points must be specified in each figure legend:

- the name of the statistical test used to generate error bars and P values,
 - the number (n) of independent experiments (please specify technical or biological replicates) underlying each data point,
 - the nature of the bars and error bars (s.d., s.e.m.)
- If the data are obtained from n {less than or equal to} 5, show the individual data points in addition to the SD or SEM.
- If the data are obtained from n {less than or equal to} 2, use scatter blots showing the individual data points.

See also the guidelines for figure legend preparation:
<https://www.embopress.org/page/journal/14693178/authorguide#figureformat>

11) Our journal encourages inclusion of *data citations in the reference list* to directly cite datasets that were re-used and obtained from public databases. Data citations in the article text are distinct from normal bibliographical citations and should directly link to the database records from which the data can be accessed. In the main text, data citations are formatted as follows: "Data ref: Smith et al, 2001" or "Data ref: NCBI Sequence Read Archive PRJNA342805, 2017". In the Reference list, data citations must be labeled with "[DATASET]". A data reference must provide the database name, accession number/identifiers and a resolvable link to the landing page from which the data can be accessed at the end of the reference. Further instructions are available at <<https://www.embopress.org/page/journal/14693178/authorguide#referencesformat>>.

12) All Materials and Methods need to be described in the main text. We would encourage you to use 'Structured Methods', our

new Methods format. According to this format, the Methods section should include a Reagents and Tools Table (listing key reagents, experimental models, software and relevant equipment and including their sources and relevant identifiers) followed by a Methods and Protocols section in which we encourage the authors to describe their methods using a step-by-step protocol format with bullet points, to facilitate the adoption of the methodologies across labs. More information on how to adhere to this format as well as downloadable templates (.doc or .xls) for the Reagents and Tools Table can be found in our author guidelines: < <https://www.embopress.org/page/journal/14693178/authorguide#manuscriptpreparation>>.

An example of a Method paper with Structured Methods can be found here:
<<https://www.embopress.org/doi/10.15252/msb.20178071>>.

13) As part of the EMBO publication's Transparent Editorial Process, EMBO Reports publishes online a Review Process File to accompany accepted manuscripts. This File will be published in conjunction with your paper and will include the referee reports, your point-by-point response and all pertinent correspondence relating to the manuscript.

Yours sincerely,

Referee #1:

In this study Martín-González and colleagues gain mechanistic insights into the interaction between the sorting nexin SNX17 and the Retriever complex using in vitro approaches. They utilized previously published structures from both low and high resolution cryo-electron microscopy and utilized recombinant proteins to generate the Retriever complex bound to SNX17. They performed both SAX analysis and AlphaFold2 predictions to predict interfaces between SNX17 and Retriever. The authors then performed a series of biophysical studies and site directed mutagenesis and specifically found that binding of cargo to SNX17 releases the C-terminal region of SNX17. This C-terminal region then interacts with the Retriever complex. Using liposomes and GUVs, the authors also show that SNX17 binding to PI3P can promote Retriever recruitment in a cargo-independent manner. Overall, this is an interesting study that makes important new contributions. However, a few conclusions of the study are not well supported by the experimental data. In addition, the findings could be strengthened by the inclusion of some cell-based studies.

Line 189: The authors refer to Fig 1J but there is not such figure. It appears that this data was not in the submitted manuscript, and thus could not be evaluated.

In Figure 2D the authors test and find that among several known SNX17 cargoes, only the viral protein L2 showed significant binding. There is a concern that L2 may bind SNX17 in a way that is distinct from the endogenous cargoes. The author should comment on this possibility. Alternatively, in Figure 2E, the authors immobilize SNX17 rather than the cargo, and it appears that they obtained reasonable binding of L2. They could try this same approach of immobilized SNX17 for some of the other cargoes.

Figure 2D and E: It is not clear from the methods or figure legend how the quantification is normalized. For example, in 2E, it looks like SNX17 protein is more abundant in the SNX17+L2+R condition. Is this considered in the quantification? The authors should explain the method. In addition, rather than a simple bar graph, the authors should show the individual data points for the different replicates.

Line 296: The authors refer to data in figure Fig S5, but this figure is missing, and it was not elsewhere in the manuscript. Thus, it wasn't possible to review some claims, for example that the C-terminal mutation SNX17-L470G does not affect L2 binding. This should be shown and quantitated.

Line 424. The authors claim that the lack of Retriever binding to GUV means that Retriever does not bind to membranes. According to the composition of the GUV, the authors show that Retriever cannot bind PI3P. It could be that Retriever binds other lipids that are not part of GUVs. The authors should clarify the conclusion of these findings.

Although the in vitro data are consistent with the author's model, adding a modest number of cell-based approaches would significantly strengthen the conclusions of the study. For example, the authors claim that in the presence of PI3P which recruits SNX17, Retriever recruitment is no longer enhanced by the presence of cargoes. It is difficult to assess this claim solely with GUV or liposomes and purified proteins, which are likely present at concentrations that are much higher than the endogenous proteins. One way to further test the importance of cargoes would be to use the SNX17 mutants that are defective in binding cargoes and test if these SNX17 mutants can still bind to membranes in the presence of endogenous, active VPS34, and whether these mutants also are also competent to recruit Retriever to cell membranes. In another type of cell-based assay, the authors could test whether over-expression of the L2 cargo in cells promotes Retriever recruitment to endosomes. These studies would provide further support for the hypothesis that cargo binding to SNX17 promotes Retriever recruitment to membranes in cells.

Referee #2:

This manuscript describes a biochemical study, informed by structural predictions, of the interaction between sorting nexin 17 (SNX17) and Retriever, a complex involved in recycling cargo from endosomes to the plasma membrane. The authors find clear evidence that a SNX17:Retriever interaction is facilitated when SNX17 binds to cargo. They propose that the regulatory mechanism underlying this phenomenon is that cargo, in binding SNX17, displaces the SNX17 C-terminus, which then binds Retriever. I think that most (but not all; see below) of the evidence that is presented supports this model. The authors also document cargo-independent activation of Retriever binding when SNX17 binds to PI3P-containing membranes. The data look pretty convincing but the mechanism underlying - and rationale for - this behavior is less clear. In any case, this study makes a nice complement to several recent structural studies of Retriever in the context of the Commander Complex. I believe that a suitably revised manuscript would be a good candidate for publication in EMBO Reports.

1. Can the authors please explain more clearly their rationale for concluding, based on the SAXS data in Fig. 1, that VPS26C is flexible but Retriever is not? The VPS26C Kratky plot, for example, looks so noisy it's hard to imagine it is definitive evidence of flexibility. (However, I am not a SAXS expert.) I also question the rationale for the statement that "this flexibility positions VPS26C as a strong candidate for interaction with other proteins". Finally, 90-degree rotations would be more informative than 180-degree rotations in Fig. 1 panels E and I.

2. The VPS26C subunit looks fine in Figs. 5, S1, and S9, but it is smeary and/or invisible in Figs. 2, 3, S5, and S8. What's going on? How might this impact the results? On a related topic, the authors paid little attention to VPS26C in their description of the predicted SNX17/Retriever interaction. Why? Does an AlphaFold model for VPS35L and SNX17 (without VPS26C) predict essentially the same interaction? Did the authors make (or consider making) mutations in VPS26C to weaken SNX17 binding? Why or why not? If they did, what happened?

3. While the data presented are largely consistent with the authors' cargo-enhanced SNX17:Retriever binding model, Figs. 4E,F raise possible red flags. The model predicts that the SNX17 H457A and N459A+F462A mutants should, by tending to release the inhibitory C-terminus from the cargo-binding site, improve the binding of cargo and SNX17CT-18. Instead, in 3 of 4 cases, it weakens binding; in the fourth case there is no significant effect. The authors need to address this complication. How do they imagine that mutations in the inhibitory C-terminus are able to weaken cargo binding, or binding of C-terminal peptides in trans?

4. As a structural biologist, I found the structure figures rather difficult to interpret. It would be helpful, in my view, to revise them with the following considerations. (1) Transparency can sometimes be useful, but it should be deployed sparingly. In particular, it shouldn't make it hard to tell whether one element is in front of or behind another. This was a major problem, for example, with Figs. 3 and S2. In most instances, I would avoid semi-transparent ribbons and/or surfaces altogether. (2) A major challenge in all structural representations is keeping the viewer oriented. If the same model is presented from multiple perspectives, the perspectives need to be presented side-by-side with the relationship (e.g., a 90-degree rotation) indicated. If related models are presented (e.g., in Fig. S2A,B), they should be shown in the same orientation. (I found Fig. S2B to be essentially uninterpretable.) If two different representations are shown of the same model from the same perspective (e.g., Figs. 2A and S2A), they need to be presented side-by-side (as in Figs. S2C,D). (3) Stick representations (e.g., Fig. 3B) are clearer when hydrogens are omitted, as is the almost universal practice. (4) In Fig. 3B, Leu 470's carboxyl group, which should display two red oxygens, appears to be missing, and is described in the text as forming a salt bridge with E248, which should be R248. (5) I was

unable to make much out of Fig. S2F, especially the top panel. The bottom right panel, labeled II, seems to be basically the same as the bottom panel of Fig. 3B (although it's hard to be sure). Please try to fix.

5. The authors find that either cargo engagement or membrane attachment can apparently activate SNX17 for Receiver binding. It would seem, therefore, that Figs. 5B and S10B would need to illustrate the state in which cargoless SNX17 recruits Receiver. The putative purpose/implications of this state merit discussion..

6. A large number of PAE matrices are shown in the supplemental figures, but as far as I could tell they were never interpreted in the text. How do these plots support (or otherwise) the inferred intermolecular (and intramolecular, for autoinhibition) interactions?

Referee #3:

The manuscript by Martin-Gonzalez and co-workers details experiments to investigate the mechanism(s) through which SNX17 interacts with the retriever complex and how the combination of binding SNX17 and cargo can mediate membrane association of the retriever complex.

The retriever complex is related to retromer and shares the VPS29 subunit with retromer. However, unlike retromer, retriever is not conserved across all eukaryotes and proteomic studies have indicated that expression of retriever is less than retromer. Some of the cargo proteins reportedly sorted by retriever are also sorted by retromer in conjunction with SNX27. For example, integrins and Glut1 are both known to be sorted by SNX17-retriever and SNX27-retromer from endosomes to the cell surface. The study by Martin-Gonzalez and co-workers is conducted in vitro and employs binding assays, some low resolution structural data and structural predictions using AlphaFold. Although the in vitro data is, in my view, generally solid, the lack of any in vivo data to support the claims/conclusions made significantly weakens the study. Additionally, much of the data obtained centers on binding of SNX17-retriever to the cytoplasmic domain of a virally encoded protein and it is not impossible that the mechanisms that control that binding are not the same as would occur for binding of endogenous cargoes to SNX17-retriever. Overall I feel that some additional experimental data is required before the study is suitable for publication.

1. None of the observations reported are backed up by experiments conducted in vivo. Mutations to key residues that abolish binding in vitro (eg L470G) could (and should) be made to the respective protein(s) and tested for an effect on binding/recruitment in cells. Without some in vivo data that corroborates the observations made in vitro, the data presented does not significantly advance the understanding of the mechanisms of SNX17-retriever function.

2. What happens if the putative NPxY motif in the L2 cytoplasmic domain is mutated? Presumably binding is abolished? Does this impact on interactions with SNX17-retriever in vitro and in vivo?

3. Can the L2 tail pulldown retriever and/or SNX17 from a cell lysate?

4. The Graph shown in Figure 2E is potentially misleading when describing the amount of VPS35L bound as the graph is normalised to a condition that gives maximal binding. It would be better to graph how much VPS35L binds relative to the input.

5. Much of the referencing of published articles could be improved. Too few of the early studies of retromer are cited.

We thank the reviewers for their thoughtful critiques. The answers to their concerns and changes made to the paper are outlined below in response to their comments.

In the revised manuscript changes in the text are highlighted. Due to the extensive reorganization of many figures, the changes in the figure codes have not been highlighted in the text to maintain clarity in the revisions made.

Here is a summary of the changes made in the figures:

Figure 2D: now is Figure EV2A.

Figure 2E: now completed with endogenous cargos in Figure 2D.

Figure 3C: now includes also the cargo LRP1.

Figure 4E, 4F and S8A: are now fused in Figure 4E. The experiments were repeated with fresh purified protein and the new calculated K_{DS} are reported.

Figure 4G: now includes also the cargo LRP1.

Figure 5C: has been replaced with the co-sedimentation assay using the LRP1 cargo (Figure S9C including more replicates). The data with cargo L2 are now in Figure EV4B. The graph type has been changed from an interleaved scatter plot with bars to a separated scatter plot with bars for better visualization.

Figure S1A-B: now in Appendix Figure S1.

Figure S1C-F: now in Figure EV1.

Figure S1G-H: now in Appendix Figure S2.

Figure S2: now in Appendix Figure S4.

Figure S3A: now in Appendix Figure S5, and the alignment of the full-length protein is depicted.

Figure S3B: now in Appendix Figure S6.

Figure S4: now in Appendix Figure S7, and the alignment of the full-length protein is depicted.

Figure S5A: now in Figure EV3A.

Figure S5B-C: now in Figure EV3B.

Figure S6: now in Appendix Figure S9.

Figure S7: now in Appendix Figure S10.

Figure S8A: now in Figure 4E.

Figure S8B: now included in Source Data for Figure 4F.

Figure S9A: now in Appendix Figure S11.

Figure S9B: now in Figure EV4A.

Figure S9C: now in Figure 5C.

Figure S9D-E: now in Figure EV4C-E.

Full new figures included in the reviewed manuscript:

Figure EV2B: Effect of NPxY motif mutations in cargos on the cargo-mediated interaction between Retriever and SNX17.

Figure EV2C: Impact of salt concentration on the SNX17-Retriever interaction in the presence or absence of cargo.

Figure EV4D: Effect of PI5P on Retriever recruitment to membranes.

Appendix Figure S2C-E. Porod-Debye plots to illustrate VPS26C protein flexibility.

Appendix Figure S3. *In vivo* pull-downs of overexpressed L2 proteins.

Appendix Figure S9: Far-UV CD spectra of wild-type MBP-SNX17 and its mutants used in this study.

Referee #1:

In this study Martín-González and colleagues gain mechanistic insights into the interaction between the sorting nexin SNX17 and the Retriever complex using *in vitro* approaches. They utilized previously published structures from both low and high resolution cryo-electron microscopy and utilized recombinant proteins to generate the Retriever complex bound to SNX17. They performed both SAX analysis and AlphaFold2 predictions to predict interfaces between SNX17 and Retriever. The authors then performed a series of biophysical studies and site directed mutagenesis and specifically found that binding of cargo to SNX17 releases the C-terminal region of SNX17. This C-terminal region then interacts with the Retriever complex. Using liposomes and GUVs, the authors also show that SNX17 binding to PI3P can promote Retriever recruitment in a cargo-independent manner. Overall, this is an interesting study that makes important new contributions. However, a few conclusions of the study are not well supported by the experimental data. In addition, the findings could be strengthened by the inclusion of some cell-based studies.

1. Line 189: The authors refer to Fig 1J but there is not such figure. It appears that this data was not in the submitted manuscript, and thus could not be evaluated.

We apologize for the error. The panel J is not included in this manuscript.

2. In Figure 2D the authors test and find that among several known SNX17 cargoes, only the viral protein L2 showed significant binding. There is a concern that L2 may bind SNX17 in a way that is distinct from the endogenous cargoes. The author should comment on this possibility. Alternatively, in Figure 2E, the authors immobilize SNX17 rather than the cargo, and it appears that they obtained reasonable binding of L2. They could try this same approach of immobilized SNX17 for some of the other cargoes.

To address the concerns regarding the extrapolation of the data obtained for L2 to endogenous cargoes, we first conducted the suggested experiment of immobilizing MBP-SNX17 and testing Retriever binding in the presence of endogenous cargoes and L2 (Fig 2D / **Figure 1** below). Consistent with previous pull-down experiments using GST-cargos (Fig. EV2A), we observed a significant increase in the Retriever-SNX17 interaction in the presence of L2 but also with the endogenous cargo LRP1, facilitated by the direct immobilization of SNX17. No significant interaction was detected in the presence of APP or ITGB1, likely due to their low affinity for SNX17, compared to that of L2 and LRP1, which may lead to their dissociation during the washing steps of the pull-down experiments. We describe this assay in the revised text (lines 235-254).

Figure 1. SNX17 interaction with cargo triggers Retriever recruitment.

The interaction of the Retriever complex with MBP-SNX17 was evaluated in the presence and absence of the cargoes LRP1_{ICD}, APP_{ICD}, ITGB1_{ICD}, and L2_{FBR}, each fused with GST, in MBP pull-down assays. Non-fused MBP protein was used as a negative control. Proteins were visualized by Coomassie Blue staining. The right panel shows the quantification of the Retriever binding to SNX17. Quantification was carried out using ImageJ, measuring VPS35L as a representative band of the Retriever complex. The ratio of the VPS35L pull-down band to the MBP-SNX17 band was calculated in each lane, assuming a one-to-one binding

stoichiometry. Non-specific binding of VPS35L to MBP was subtracted from the VPS35L band intensities. The results are expressed as mean \pm SD (n = 4 technical replicates). Statistical analysis was performed using unpaired Student's t-test, with cargo vs. without cargo. ** p = 0.004.

Given that LRP1 cargo enables *in vitro* observation of the SNX17-Retrieve association, we repeated several experiments using LRP1 instead of L2. We used LRP1 in the pull-down assays that demonstrate that the C-terminus of SNX17 directly interacts with Retriever (Fig. 3C / **Figure 2** below) and in the assays that show that the autoinhibition mechanism is released by cargo binding (Fig. 4F / **Figure 3** below). In these assays we obtained equivalent results with L2 and LRP1, indicating that cargo-mediated activation occurs with endogenous cargos, and that the HPV virus, through its L2 protein, can recruit Retriever using the same mechanism.

Figure 2. The C-terminal end of SNX17 contacts the VPS35L/VPS26C interface.

Analysis of the interaction between Retriever and SNX17 mutants in the presence and absence of the cargos LRP1 or L2. MBP pull-down assays were performed with wild-type MBP-SNX17 or indicated mutants, Retriever, GST-LRP1_{ICD} or GST-L2_{FBR}. Non-fused MBP protein was used as a negative control. Samples were loaded onto an SDS-PAGE gel and stained with Coomassie Blue. Quantification was carried out as detailed in Fig. 2D. The graph represents the mean \pm SD of technical replicates (LRP1: n = 3; L2: n=2).

Figure 3. SNX17 autoinhibition mechanism for Retriever binding in the absence of cargo.

Purified Retriever complex was incubated with the indicated MBP-SNX17 mutants in the presence or absence of GST-LRP1_{ICD} or GST-L2_{FBR} in MBP pull-down assays. Non-fused MBP protein was used as a negative control. Quantification of the Coomassie stained SDS-PAGE gel was carried out in ImageJ, measuring VPS35L as a representative band of the Retriever complex. The level of Retriever binding to MBP-SNX17 was quantified as described in Fig. 2D. Values represent mean \pm SD (n = 2 biological replicates, with MBP-SNX17 and its mutants obtained from two independent protein purifications). R, retriever; C, cargo.

3. Figure 2D and E: It is not clear from the methods or figure legend how the quantification is normalized. For example, in 2E, it looks like SNX17 protein is more abundant in the SNX17+L2+R condition. Is this considered in the quantification? The authors should explain the method. In addition, rather than a simple bar graph, the authors should show the individual data points for the different replicates.

Following the reviewer's suggestion, we have revised the quantification method used for all the pull-down assays presented in this study. The method of quantification is now described in detail in all figure legends. Additionally, the Source Data provided with this resubmission, includes an Excel file for each pull-down quantification, with a description of the method used in the first tab of each file.

In Figure 2D (now Fig. EV2A) the band intensities of SNX17 and VPS35L were normalized to the GST or GST-cargo band intensity to account for slight differences in sample loading. Non-specific binding to GST was subtracted from the normalized bands. The percentage of SNX17 or VPS35L binding to GST-cargos was calculated as the ratio of the pull-down protein to the input protein (lanes 2 and 3). The samples were loaded to ensure that input and resuspended beads contained proportionally the same amount.

In the Figure 2E (now Fig. 2D) the ratio of the VPS35L pull-down band to the MBP-SNX17 band was calculated in each lane, assuming a one-to-one binding stoichiometry. Non-specific binding of VPS35L to MBP was subtracted from the VPS35L band intensities. We could calculate this ratio for quantifying the percentage of binding because MBP-SNX17 and VPS35L have similar size and bind the Coomassie dye with similar intensities when the same amount is loaded in a gel (see input lanes in gels of Figs EV2B and C).

As the reviewer also suggested, we have now included the individual data points in the quantification of pull-downs and co-sedimentation assays. The data points are color-coded according to the experiment from which they originate. This approach clarifies the trends observed in some experiments, which were not as apparent when only the standard deviation was shown.

4. Line 296: The authors refer to data in figure Fig S5, but this figure is missing, and it was not elsewhere in the manuscript. Thus, it wasn't possible to review some claims, for example that the C-terminal mutation SNX17-L470G does not affect L2 binding. This should be shown and quantitated.

We apologize for the error in the initial submission of the article, where Figure S5 was mistakenly omitted. In this revised version, Figure S5 (now Fig. EV3) has been correctly included.

5. Line 424. The authors claim that the lack of Retriever binding to GUV means that Retriever does not bind to membranes. According to the composition of the GUV, the authors show that Retriever cannot bind PI3P. It could be that Retriever binds other lipids that are not part of GUVs. The authors should clarify the conclusion of these findings.

Thank you for pointing this out, the reviewer is correct, and this conclusion was indeed overstated. We have removed the conclusion from line 469: "In contrast, when mKate2-Retriever was incubated with GUVs, no protein was detected bound to the GUV membranes, indicating that Retriever lacks membrane binding ability". Additionally, in the Discussion section, line 628, we have added: "Whether other lipid compositions influence the recruitment of the Retriever complex to the membrane requires further investigation".

For this revision, we have further investigated whether phosphatidylinositol 5-phosphate (PI5P) could serve as ligand for Retriever recruitment to membranes. However, in our binding assays with liposomes, no interaction was observed (Fig EV4D). We also repeated the co-sedimentation assays using the Folch-fraction I from bovine brain as the lipid source, which contains phospholipids, triglycerides, cholesterol, and other lipid types. In these assays Retriever was not found to bind significantly to the lipidic fraction after co-sedimentation (**Figure 4**, below). We did not include these results in our revised manuscript, because phosphoinositols are found in a low amount in Folch Fraction I. SNX17 co-sedimented with liposomes produced with the Folch fraction I and this hinders our ability to accurately study the effect of adding PI3P to the lipidic samples.

Figure 4. Effect of lipid composition on Retriever recruitment liposomes.

Co-sedimentation assay performed by incubating liposomes produced from the Folch I fraction of bovine extract, either alone or supplemented with PI3P.

6. Although the *in vitro* data are consistent with the author's model, adding a modest number of cell-based approaches would significantly strengthen the conclusions of the study. For example, the authors claim that in the presence of PI3P which recruits SNX17, Retriever recruitment is no longer enhanced by the presence of cargoes. It is difficult to assess this claim solely with GUV or liposomes and purified proteins, which are likely present at concentrations that are much higher than the endogenous proteins. One way to further test the importance of cargoes would be to use the SNX17 mutants that are defective in binding cargoes and test if these SNX17 mutants can still bind to membranes in the presence of endogenous, active VPS34, and whether these mutants also are also competent to recruit Retriever to cell membranes. In another type of cell-based assay, the authors could test whether over-expression of the L2 cargo in cells promotes Retriever recruitment to endosomes. These studies would provide further support for the hypothesis that cargo binding to SNX17 promotes Retriever recruitment to membranes in cells.

In the initial version of this manuscript, there was a slight increase in Retriever recruitment to PI3P-liposomes with SNX17 in the presence of cargo. However, due to variability between experiments, this increase was not statistically significant. For this revision, we increased the number of replicates in the co-sedimentation assays from 3 to 6 (Fig. 5C), and this time, a significant increase in Retriever recruitment was observed, which has been discussed in the text (lines 498-502).

The cellular experiments suggested by the reviewer are highly relevant, and we agree that they would further strengthen our proposed mechanism based on *in vitro* experiments. However, we consider that the first proposed experiment, -investigating whether SNX17 mutants that are defective in cargo binding are capable of recruiting Retriever to membranes-, may not yield conclusive results. Cellular studies have shown that SNX17 requires both cargo and PI3P recognition for endosomal recruitment. For example, SNX17 mutants deficient in cargo binding, such as W321A, have been previously studied in CHO-InsR cells (Ghai et al, 2013). By immunofluorescence labelling was observed that this SNX17 mutant lost its punctuate endosomal recruitment and exhibited a diffuse cytoplasmic distribution. The requirement of PI3P was proven in cellular assays through two different approaches. First, the R36Q

mutant of the PX domain, which does not bind PI3P, shows only cytosolic localization (Ghai et al., 2011). Second, inhibition of the PI3 kinases activity with wortmannin caused SNX17 punctuate localization on endosomes to shift to a cytosolic distribution (Tseng et al, 2014). Consequently, cargo-defective mutants would likely not bind to membranes, making it difficult to test their ability to recruit Retriever on endosomes without cargo binding.

As suggested by the reviewer, we have incorporated cell-based experiments into our revised manuscript to investigate the role of cargo in promoting Retriever recruitment. We overexpressed the L2 cargo in HEK293T cells, using both the short construct purified in this study (L2_{FBR}) and the full-length protein. Subsequent pull-down assays with overexpressed L2 in HEK293 cells demonstrated the effective recruitment of SNX17 and Retriever (Appendix Fig. S3). Further details of these results are provided in our response to point 3 of Referee 3.

Referee #2:

This manuscript describes a biochemical study, informed by structural predictions, of the interaction between sorting nexin 17 (SNX17) and Retriever, a complex involved in recycling cargo from endosomes to the plasma membrane. The authors find clear evidence that a SNX17:Retriever interaction is facilitated when SNX17 binds to cargo. They propose that the regulatory mechanism underlying this phenomenon is that cargo, in binding SNX17, displaces the SNX17 C-terminus, which then binds Retriever. I think that most (but not all; see below) of the evidence that is presented supports this model. The authors also document cargo-independent activation of Retriever binding when SNX17 binds to PI3P-containing membranes. The data look pretty convincing but the mechanism underlying - and rationale for - this behavior is less clear. In any case, this study makes a nice complement to several recent structural studies of Retriever in the context of the Commander Complex. I believe that a suitably revised manuscript would be a good candidate for publication in EMBO Reports.

1. Can the authors please explain more clearly their rationale for concluding, based on the SAXS data in Fig. 1, that VPS26C is flexible but Retriever is not? The VPS26C Kratky plot, for example, looks so noisy it's hard to imagine it is definitive evidence of flexibility. (However, I am not a SAXS expert.) I also question the rationale for the statement that "this flexibility positions VPS26C as a strong candidate for interaction with other proteins". Finally, 90-degree rotations would be more informative than 180-degree rotations in Fig. 1 panels E and I.

The reviewer's concerns regarding the interpretation of SAXS data are acknowledged. Due to space limitations, the detailed SAXS analysis was not included in the initial manuscript. The Kratky plot is a standard tool for estimating protein flexibility. For highly flexible proteins, the Kratky plot can be less well-defined and may exhibit noise at higher q values, reflecting genuine structural variability within the sample rather than just experimental noise. To further characterize and compare the protein flexibility of VPS26C and Retriever we calculated Porod-Debye plots and incorporated them into Appendix Figure S2. The analysis process involves comparing the scattering curves of the particles under study over a range of points initially determined by Guinier analysis and identifying an asymptotic region for each of them, as described by Rambo and Tainer (2011). The plateau region in the $q^4 I(q)$ vs. q^4 plots is indicative of less flexible particles, in the $q^3 I(q)$ vs. q^3 plots for partially flexible proteins, and in the $q^2 I(q)$ vs. q^2 plots for highly flexible particles. To perform these analyses, we used the BioXTAS RAW software and the SASPLOT function incorporated into the ATSAS software (Figure 5 / Appendix Fig. S2). Our analysis shows that the scattering profile of the Retriever complex displays an asymptotic region in the $q^4 I(q)$ vs. q^4 plot, while it is not yet visible for VPS26C. When comparing with the $q^3 I(q)$ vs. q^3 plot, the loss of the plateau region for the Retriever complex is observed, and the asymptotic trend of the VPS26C curve begins to emerge, a feature that becomes fully apparent in the $q^2 I(q)$ vs. q^2 plot. These results clearly demonstrate the pronounced flexibility of VPS26C in contrast to the compact behaviour of the Retriever complex.

We propose in the manuscript that "this flexibility positions VPS26C as a strong candidate for interaction with other proteins" because flexible proteins can undergo conformational changes that increase their potential for interacting with a variety of other proteins (Marsh et al 2012). To address the reviewer's suggestion, we have reformulated the statement to improve clarity: "This flexibility may facilitate its interaction with VPS35L for the formation of the Retriever complex and potentially enable interactions of Retriever with other partners" (lines 180-182).

As requested by the reviewer, we have included 90-degree rotation images in Fig. 1E and Fig. 1I, which provide a better visualization of the fitting of the VPS26C and Retriever models within the *ab initio* SAXS envelope.

Furthermore, we have deposited the SAXS data in the curated repository SASBDB database.

Figure 5. Porod-Debye plots showing VPS26C as a highly flexible particle. Comparison of the changes in the Porod-Debye regions between $q^4 I(q)$ vs. q^4 (A), $q^3 I(q)$ vs. q^3 (B), $q^2 I(q)$ vs. q^2 (C) plots for both VPS26C and the Retriever complex.

2. The VPS26C subunit looks fine in Figs. 5, S1, and S9, but it is smeary and/or invisible in Figs. 2, 3, S5, and S8. What's going on? How might this impact the results?

As correctly noted by the reviewer, the VPS26C protein exhibits a smeared appearance on the indicated gels. This anomalous electrophoretic behavior is consistently observed when using commercial gradient gels (precast Invitrogen NuPAGE 4–12% Bis-Tris SDS-PAGE gels). However, when loading the same sample onto our self-made 12% or 15% Tris SDS-PAGE gels, it appears as a well-defined band. Given that the loading buffer remains constant, this discrepancy can be attributed to differences in the composition of the gels or the electrophoresis buffer. Nevertheless, we can confidently assert that this phenomenon is not due to protein degradation, as a stable, non-degraded complex is observed running the same sample in self-made gels. For quantification, we selected the VPS35L band from the Retriever complex rather than the VPS26C band, thus this artifact did not affect our results.

On a related topic, the authors paid little attention to VPS26C in their description of the predicted SNX17/Retriever interaction. Why? Does an AlphaFold model for VPS35L and SNX17 (without VPS26C) predict essentially the same interaction? Did the authors make (or consider making) mutations in VPS26C to weaken SNX17 binding? Why or why not? If they did, what happened?

In AlphaFold2-generated models, the C-terminus of SNX17 consistently resides at the interface between VPS26C and VPS35L, forming specific side-chain interactions with residues of VPS35L. However, specific interactions between VPS26C residues and the SNX17 CT-18 peptide were not consistently observed across all structures. Therefore, site-directed mutagenesis of VPS26C residues was not performed to assess their role in the interaction.

As suggested by the reviewer, we also explored whether AlphaFold could predict the interaction between the C-terminal region of SNX17 and either VPS26C or VPS35L individually (**Figure 6**). Analysis of models generated by AlphaFold2 revealed that the observed interaction of SNX17-CT18 with the VPS26C+VPS35L₁₁₀₋₅₉₈ subcomplex, was correctly predicted even when only the VPS35L subunit was used, but not with VPS26C. Similar results were obtained with AlphaFold3.

However, our fluorescence anisotropy data indicate that the SNX17 C-terminal peptide requires the formation of the VPS26C-VPS35L complex for stable interaction. While the affinity of VPS35L+VPS29 alone is low, the VPS26C+VPS35L₁₁₀₋₅₉₈ complex exhibits an affinity comparable to that of the full complex, suggesting a crucial role for VPS26C in the interaction with SNX17.

Figure 6. AlphaFold models of SNX17 with the individual subunits of the Retriever complex.

AlphaFold models of SNX17-CT18 with the Retriever subcomplex VPS26C+VPS35L₁₁₀₋₅₉₈ (A), and the individual subunits VPS26C (B) and VPS35L (C).

3. While the data presented are largely consistent with the authors' cargo-enhanced SNX17:Retriever binding model, Figs. 4E,F raise possible red flags. The model predicts that the SNX17 H457A and N459A+F462A mutants should, by tending to release the inhibitory C-terminus from the cargo-binding site, improve the binding of cargo and SNX17CT-18. Instead, in 3 of 4 cases, it weakens binding; in the fourth case there is no significant effect. The authors need to address this complication. How do they imagine that mutations in the inhibitory C-terminus are able to weaken cargo binding, or binding of C-terminal peptides in trans?

We thank the reviewer for bringing this issue to our attention. Addressing the concerns raised, we have strengthened the evidence supporting our proposed mechanism in the revised manuscript. The purification of SNX17 expressed in bacteria yielded moderate results and low solubility. Therefore, we opted for MBP-SNX17 constructs, which increase the protein's solubility and allow us to reach the concentrations required to calculate dissociation constants in fluorescence anisotropy assays. Despite implementing various strategies to minimize protein degradation, including the addition of protease inhibitors and an extra ion-exchange chromatography step, minor degradation was still observed, especially in the mutant proteins. We checked the degree of degradation of the SNX17 mutants H457A and N459A+F462A used in the fluorescence anisotropy assays from the first version of the manuscript and found it to be slightly higher than that of the wild-type protein. As a result, the protein concentration estimates were not accurate and neither K_D s. To ensure accurate protein concentration determination, we have repurified the mutants and calibrated the nanodrop measurements against Coomassie-stained gel intensities (Figure 7). Circular dichroism spectroscopy confirmed that the mutations did not compromise protein folding (Figure 8, Appendix Fig. S8). Subsequent fluorescence anisotropy experiments, presented in the revised Figure 4E, revealed that the N459A+F462A mutant, which disrupts the autoinhibitory conformation, exhibits a higher affinity for the cargo, as anticipated based on our model. The explanation for this section has been updated according to the new data in lines 419-424 of the manuscript.

Figure 7. Protein concentration adjustment of MBP-SNX17 mutants accounting for protein degradation. Coomassie-stained SDS-PAGE gels with the same amount of protein loaded based on nanodrop A280 nm measurement (A) or after adjustment of protein concentration based on band intensity quantification of gel in A (B).

Figure 8. Conformational properties of MBP-SNX17 mutants. Far-UV CD spectra of WT MBP-SNX17 and its mutants used in this work. MRE, mean residual ellipticity.

4. As a structural biologist, I found the structure figures rather difficult to interpret. It would be helpful, in my view, to revise them with the following considerations. (1) Transparency can sometimes be useful, but it should be deployed sparingly. In particular, it shouldn't make it hard to tell whether one element is in front of or behind another. This was a major problem, for example, with Figs. 3 and S2. In most instances, I would avoid semi-transparent ribbons and/or surfaces altogether. (2) A major challenge in all structural representations is keeping the viewer oriented. If the same model is presented from multiple perspectives, the perspectives need to be presented side-by-side with the relationship (e.g., a 90-degree rotation) indicated. If related models are presented (e.g., in Fig. S2A,B), they should be shown in the same orientation. (I found Fig. S2B to be essentially uninterpretable.) If two different representations are shown of the same model from the same perspective (e.g., Figs. 2A and S2A), they need to be presented side-by-side (as in Figs. S2C,D). (3) Stick representations (e.g., Fig. 3B) are clearer when hydrogens

are omitted, as is the almost universal practice. (4) In Fig. 3B, Leu 470's carboxyl group, which should display two red oxygens, appears to be missing, and is described in the text as forming a salt bridge with E248, which should be R248. (5) I was unable to make much out of Fig. S2F, especially the top panel. The bottom right panel, labeled II, seems to be basically the same as the bottom panel of Fig. 3B (although it's hard to be sure). Please try to fix.

We are grateful to the reviewer for his/her meticulous analysis of the structural images. We have carefully implemented most of the suggestions, resulting in significantly improved figure clarity and informativeness. We have tried to maintain the same orientation in all models of Appendix Fig. 4. However, a different orientation was used in panel A to properly visualize the SNX17-Retrieve interaction. Additionally, we have included the PDB and JSON files in the Source Data to facilitate sharing our models with the broader scientific community and enable further analysis.

5. The authors find that either cargo engagement or membrane attachment can apparently activate SNX17 for Retriever binding. It would seem, therefore, that Figs. 5B and S10B would need to illustrate the state in which cargoless SNX17 recruits Retriever. The putative purpose/implications of this state merit discussion...

Thanks for the suggestion. We have revised Fig5B and Fig EV5 to provide a more accurate illustration of the membrane-mediated activation mechanism. The corresponding Figure legend E5V has been updated to reflect these changes. Our findings suggest that the activation mechanism through specific binding to PI3P-enriched membranes acts as a spatiotemporal control, ensuring that SNX17 is only active at endosomal membranes. This induced conformational change may also contribute to enhancing cargo binding by liberating the cargo binding pocket.

We have incorporated a discussion of these implications into the manuscript, specifically in lines 520-526.

6. A large number of PAE matrices are shown in the supplemental figures, but as far as I could tell they were never interpreted in the text. How do these plots support (or otherwise) the inferred intermolecular (and intramolecular, for autoinhibition) interactions?

Thank you for pointing this out. We agree that PAE matrices play a significant role in interpreting structural models, as they provide valuable insights into the confidence level of the predictions. Interestingly, the position of the last 18 residues of SNX17 (CT-18) relative to the Retriever complex exhibits high confidence in all analyzed structures, supporting our experimental data (S4A-C). However, the predicted position of the remaining portion of the SNX17 model relative to the Retriever complex shows lower confidence, particularly when modeling the entire Retriever complex (Appendix Fig. S4A) When focusing on the interaction with the VPS26C:VPS35L₁₁₀₋₅₉₈ complex, the confidence level improves to moderate. These PAE values underscore the flexibility of the long C-terminal region of SNX17 and further substantiate our proposed model. We have incorporated a discussion of these findings into the revised manuscript, specifically in lines 319-322.

Furthermore, the PAE values are consistent with our proposed autoinhibition mechanism. In the AlphaFold2 models of human SNX17 and its orthologs, the position of the C-terminus consistently displays low predicted errors (dark green) (Appendix Figure S9). However, residues 398-452 in human SNX17 and their equivalent residues in orthologs exhibit high predicted errors (white or light green), indicating the flexibility of this region. We have clarified these observations in the revised manuscript (lines 394-395).

Additionally, in the modeled complexes of SNX17 with cargo peptides, the position of the cargo is well-supported by the PAE matrices.

Referee #3:

The manuscript by Martin-Gonzalez and co-workers details experiments to investigate the mechanism(s) through which SNX17 interacts with the retriever complex and how the combination of binding SNX17 and cargo can mediate membrane association of the retriever complex.

The retriever complex is related to retromer and shares the VPS29 subunit with retromer. However, unlike retromer, retriever is not conserved across all eukaryotes and proteomic studies have indicated that expression of retriever is less than retromer.

Some of the cargo proteins reportedly sorted by retriever are also sorted by retromer in conjunction with SNX27. For example, integrins and Glut1 are both known to be sorted by SNX17-retriever and SNX27-retromer from endosomes to the cell surface.

The study by Martin-Gonzalez and co-workers is conducted *in vitro* and employs binding assays, some low resolution structural data and structural predictions using AlphaFold. Although the *in vitro* data is, in my view, generally solid, the lack of any *in vivo* data to support the claims/conclusions made significantly weakens the study. Additionally, much of the data obtained centers on binding of SNX17-retriever to the cytoplasmic domain of a virally encoded protein and it is not impossible that the mechanisms that control that binding are not the same as would occur for binding of endogenous cargoes to SNX17-retriever.

Overall I feel that some additional experimental data is required before the study is suitable for publication.

We fully agree with the reviewer that, since our initial version primarily used the viral L2 protein, the relevance of the proposed mechanism for endogenous cargoes could be questioned.

To address this concern, we have employed the endogenous cargo LRP1 in a series of experiments to validate the mechanism. These new experiments, detailed in our response to Reviewer 1, comment 1, have confirmed that the proposed mechanism is indeed valid with endogenous cargoes and is not an exclusive process induced by the binding of the viral HPV protein L2. We have reviewed the manuscript accordingly.

1. None of the observations reported are backed up by experiments conducted *in vivo*. Mutations to key residues that abolish binding *in vitro* (eg L470G) could (and should) be made to the respective protein(s) and tested for an effect on binding/recruitment in cells. Without some *in vivo* data that corroborates the observations made *in vitro*, the data presented does not significantly advance the understanding of the mechanisms of SNX17-retriever function.

The cellular assays suggested by the reviewer have been previously reported in the literature and were therefore not replicated in this study. The SNX17 L470G and D467X mutants were specifically chosen for their prior use in cellular assays (McNally et al., 2017), as referenced in lines 337-339 of our manuscript. Previous GFP-trap assays using cell lysates expressing these mutants demonstrated that while they could retain the cargo LRP1, they failed to bind the Retriever complex or recruit the CCC complex proteins CCDC93 and CCDC22. A limitation of these *in vivo* assays was their inability to definitively establish a direct interaction between the C-terminus of SNX17 and VPS35L. Consequently, to fully elucidate the molecular details of this interaction, it was essential to conduct the *in vitro* experiments with purified proteins as presented in this study. As the reviewer correctly points out, the complementation of *in vitro* and *in vivo* published data strengthens the proposed mechanism.

2. What happens if the putative NPxY motif in the L2 cytoplasmic domain is mutated? Presumably binding is abolished? Does this impact on interactions with SNX17-retriever *in vitro* and *in vivo*?

We thank the reviewer for suggesting these experiments, which we have conducted both *in vitro* and *in vivo*, providing robust support for our proposed mechanism. We introduced the NPxY to APxA mutation

into both the L2 and LRP1 proteins. These mutations abolished the L2-SNX17 interaction *in vitro* and prevented cargo-mediated recruitment of Retriever (**Figure 9** below/ Fig. EV2B). This demonstrates that binding of the cargo to the canonical site, rather than a potential secondary binding site, is essential for the conformational change in SNX17 that promotes Retriever recruitment. These new results are explained on **lines 249-251** of the revised manuscript. *In vivo* experiments were also performed and are detailed in the response to the following point.

Figure 9. Cargo-dependent interaction of SNX17 with Retriever. The effect of mutating the conserved NPXY motif to APxA in LRP1 and L2 on the cargo-dependent Retriever-SNX17 interaction. Coomassie-stained SDS-PAGE gel of pull-down assays with MBP-SNX17 and Retriever in the presence of GST-LRP1_{ICD}, GST-LRP1_{ICD-mut} (N4470A+Y4473A), GST-L2_{FBR}, and GST-L2_{FBR-mut} (N254A+Y257A).

3. Can the L2 tail pulldown Retriever and/or SNX17 from a cell lysate?

Previous studies have investigated the colocalization of HPV16 pseudovirions with SNX17 (Bergant *et al*, 2012) and Retriever (Pim *et al*, 2021), but no published reports have examined the interaction of the individual full-length L2 protein. We agree with the reviewer that this assay is crucial for demonstrating the role of the *in vitro* observed interaction in the *in vivo* transport of the L2 protein during HPV virus infection. To address this, we collaborated with cellular biology experts Juan Carlos Acosta and Andrea Quintanilla from our institute. We examined the effects of both full-length L2 and a 30-amino acid L2 fragment containing the NPXY motif, used in our *in vitro* assays. Additionally, the N254A+Y257A mutant of these constructs were generated. These constructs were overexpressed in HEK293T cells using the pMSCV-FlagS plasmid and analyzed by anti-Flag agarose beads pull-down assays. Our results (**Figure 10/Appendix Fig. S3**) show that SNX17 is recruited by both the full-length and the smaller L2 construct. However, mutating the ²⁵⁴NPAY²⁵⁷ motif only prevented SNX17 recruitment by the smaller fragment, not the full-length protein. This is likely due to the presence of two additional potential SNX17 binding motifs in the L2 sequence of HPV16, ¹⁶⁰NPTF¹⁶³ and ³⁶³NGLY³⁶⁶. We also detected VPS35L in the proteins retained by the anti-Flag resin with L2, indicating that, like physiological cargos, L2 can recruit SNX17 and Retriever. With the smaller construct, Retriever recruitment was not evident, possibly due to the limited amount of retained SNX17. However, these assays do not exclude the possibility of a direct interaction between Retriever and L2 via a region outside the L2-FBR construct. We believe that a more in-depth investigation of this mechanism would be suitable for a separate publication focusing on the hijacking of HPV viral particles of the intracellular transport of Retriever.

Figure 10. Detection of VPS35L and SNX17 following Flag pull-down of overexpressed L2 proteins.

(A) HEK293T cells expressing either the wild-type or the mutant version (lacking the NPxY motif) of the Flag-tagged L2_{FBR} or L2_{FL} protein were lysed. The supernatants were incubated with M2 anti-Flag beads, followed by pull-down and immunoblot analysis using specific antibodies against VPS35L and SNX17. Cells expressing the empty vector served as a negative pull-down control. Actin was used as loading control for the input samples. Due to non-specific detection by the anti-Flag antibody, mRNA expression levels of the transfected plasmids were quantified by RT-PCR.

(B) Relative transcript expression levels of the GFP vector control, wild-type, and mutant versions (lacking the NPxY motif) of the L2_{FBR} or L2_{FL} proteins were measured by quantitative reverse transcription PCR (RT-qPCR).

4. The Graph shown in Figure 2E is potentially misleading when describing the amount of VPS35L bound as the graph is normalised to a condition that gives maximal binding. It would be better to graph how much VPS35L binds relative to the input.

We acknowledge the reviewer's concern regarding the potential ambiguity introduced by normalizing the data to the maximum value. To address this, we have implemented alternative quantification methods in all the pull-down quantifications. A detailed explanation of these changes can be found in our response to Reviewer 1, comment 3.

5. Much of the referencing of published articles could be improved. Too few of the early studies of retromer are cited.

In accordance with the reviewer's recommendations, citations to the pioneering work of the Retromer research group have been incorporated into the introduction to provide a broader context for our study, references (Seaman *et al*, 1997, 1998) in line 64. An additional reference to the SNX27-Retromer work was added: Simonetti *et al* 2022 in line 76.

In addition to the changes suggested by the reviewers, we have included a figure illustrating the effect of salt concentration on cargo-mediated SNX17-Retromer interactions *in vitro*. During the review process, a preprint was deposited on BioRxiv that also investigated the formation of the SNX17-Retromer complex (Singla *et al*, 2024). Their results initially appeared to contradict ours, as they

observed a high level of interaction between the two proteins in the absence of cargo. However, we have found that using our purified proteins and a salt concentration of 50 mM NaCl, as in their study, we also observe a high level of interaction in the absence of cargo, but this interaction is disrupted at higher salt concentrations. A potential explanation for this discrepancy is provided in lines 267-274. "At low salt concentrations (50 mM NaCl), a strong interaction was observed even in the absence of cargo, likely due to non-physiological interactions between oppositely charged regions. However, increasing the salt concentration to physiological levels (150 mM) significantly reduced the binding of Retriever to SNX17 in the absence of cargo, with a more pronounced effect at 300 mM NaCl (Fig. EV2C). Based on these observations, subsequent *in vitro* assays were conducted at a salt concentration of 200-300 mM NaCl, which is closer to physiological conditions and optimal for these studies.

References (not present in the manuscript):

Marsh JA, Teichmann SA, Forman-Kay JD. Probing the diverse landscape of protein flexibility and binding. *Curr Opin Struct Biol.* 2012 Oct;22(5):643-50. doi: 10.1016/j.sbi.2012.08.008.

Rambo RP, Tainer JA. Characterizing flexible and intrinsically unstructured biological macromolecules by SAS using the Porod-Debye law. *Biopolymers.* 2011 Aug;95(8):559-71. doi: 10.1002/bip.21638.

Singla A, Boesch DJ, Joyce Fung HY, Ngoka C, Enriquez AS, Song R, Kramer DA, Han Y, Juneja P, Billadeau DD, Bai X, Chen Z, Turer EE, Burstein E, Chen B. Structural basis for Retriever-SNX17 assembly and endosomal sorting. *bioRxiv [Preprint].* 2024 Mar 13:2024.03.12.584676. doi: 10.1101/2024.03.12.584676.

Dear Dr. Lucas

Thank you for the submission of your revised manuscript to EMBO reports. We have now received the full set of referee reports that is copied below.

As you will see, all referees are very positive about the study and support publication pending an extended discussion of your findings and those published by Singla et al and Butkovic et al.

From the editorial side, there are also a few things that we need before we can proceed with the official acceptance of your study:

- Your manuscript will be published in our Reports section, for which the Results and Discussion need to be combined. The revised manuscript should not exceed 27,000 characters (including spaces but excluding materials & methods and references). Please contact me if you want to discuss these formatting and character limit requirements and e.g., the need for an extended discussion (see above) further.

- Please provide up to 5 keywords.

- Please provide a 'Disclosure and competing interests statement'. For more information see <https://www.embopress.org/page/journal/14693178/authorguide#conflictsofinterest>

- Please note that as per our editorial policies all conclusions must be supported by data that are included in the manuscript, i.e., we do not allow "data not shown". With this in mind, we note that you state to have found no interaction between SNX17 and VPS26C (data not shown, page 12, line 372). Please include the relevant experiments/results that support this statement in the manuscript, e.g., the Appendix.

- Please note that the funding information in the manuscript text and the online manuscript tracking system must be congruent. We note that the following funders listed in the Acknowledgment section are missing in the online system and kindly ask you to add them:

- funding by MCIN/AEI/10.13039/501100011033 and by "ESF Investing in your future" (RYC-2016-20342) and (PRE2019-088459)

- Fundación Biofísica Bizkaia, the Basque Excellence Research Centre (BERC) program, and IT1745-22 of the Basque Government

- the Basque Government predoctoral program (PRE_2023_1_0100)

- Banco Santander

- Please remove the Reagents and Tools table from the manuscript and upload it as a separate file (file type Reagent table).

- Citations of preprints need to be labeled as follows:

A) in the text as (preprint: Laulumaa et al., 2023)

B) in the reference list by adding [PREPRINT] at the end of the reference

- Appendix:

a) Please provide the Appendix as a clean PDF file, i.e., without track changes.

b) Appendix Figure S3B: please provide information on 'n'

- Source data upload: for the main figures we need separate folders for each of the figures, i.e., one folder per figure. The Source Data for the EV figures and the Appendix should be zipped all together into one folder.

- Character count information should be removed from the title page

- MATERIALS AND METHODS should be METHODS

- Our production/data editors have asked you to clarify several points in the figure legends (see below). Please incorporate these changes in the manuscript and return the revised file with tracked changes with your final manuscript submission.

A) Figure legend text:

- Please note that the legends for figures 1c-h are not provided in the sequential manner (legend for figure 1f, g, h is provided before legend of figure 1c-e, 1d-e, 1e; respectively). This needs to be rectified.

B) Statistical test information. Only p-values that are actually shown in the figure panel(s) should (and must) be defined in the legends, all others should be removed from (or added to) the legend. Moreover, we ask for the specification of exact p-values:

- Please note that the exact p values are not provided in the legends of figures EV 2c; EV 4b.
- FigureEV2C and EV4B and other figures where relevant: please do not perform statistical analysis on results from technical replicates.

C) Replicates and error bars:

- Please note that the error bars are not defined in the legend of figure EV2C.
 - Please note that the measure of center for the error bars needs to be defined in the legend of Figure 4E.
- Just a short reminder not to forget to remove the reviewer access from the Data availability section.

- Finally, EMBO Reports papers are accompanied online by

A) a short (1-2 sentences) summary of the findings and their significance,

B) 2-3 bullet points highlighting key results and

C) a schematic summary figure that provides a sketch of the major findings (not a data image).

Please provide the summary figure as a separate file in PNG or JPG format at a size of 550x300-600 pixels (width x height).

Please note that the size is rather small and that text needs to be readable at the final size. Please send us this information along with the revised manuscript.

- On a different note, I would like to alert you that EMBO Press offers a new format for a video-synopsis of work published with us, which essentially is a short, author-generated film explaining the core findings in hand drawings, and, as we believe, can be very useful to increase visibility of the work. This has proven to offer a nice opportunity for exposure i.p. for the first author(s) of the study. Please see the following link for representative examples and their integration into the article web page:

<https://www.embopress.org/doi/full/10.15252/emj.2019103932>

With kind regards,

=====

Referee #1:

The authors adequately addressed many of the reviewers comments.

However, subsequent to the first submission, two relevant papers appeared that the authors need to discuss in this revised manuscript.

Even with these three concurrent papers, the mechanisms whereby SNX17 and Retriever are recruited to membranes and coordinate with each other to promote cargo recycling are not yet fully understood. To make this current paper useful for the field, it is essential that the authors discuss their findings in the context of the other two papers. Places where the findings agree, or have discrepancies, and places where the current paper are unique all should be pointed out.

The two papers are

Singla A, Boesch DJ, Joyce Fung HY, Ngoka C, Enriquez AS, Song R, Kramer DA, Han Y, Juneja P, Billadeau DD, Bai X, Chen Z, Turer EE, Burstein E, Chen B. Structural basis for Retriever-SNX17 assembly and endosomal sorting. bioRxiv [Preprint]. 2024 Mar 13:2024.03.12.584676. doi: 10.1101/2024.03.12.584676.

and

Butkovic R, et al. Mechanism and regulation of cargo entry into the Commander endosomal recycling pathway, Nat Commun. 2024 Aug 21;15(1):7180. doi: 10.1038/s41467-024-509710. PMID: 39168982

Singla et al. was discussed in the comments to the reviewers and should also be included in the discussion of this manuscript. While it is currently a preprint, readers of Martin-González et al. will have access to Singla et al. In addition, Butkovic et al. should be fully discussed in this current manuscript.

Referee #2:

I am satisfied by the revisions made by the authors and can now recommend acceptance.

Referee #3:

Having looked over the responses to the comments made originally and the additional data added to the manuscript, I feel that this study is now suitable for publication in EMBO Reports.

All editorial and formatting issues were resolved by the authors.

Dr. María Lucas
Instituto de Biomedicina y Biotecnología de Cantabria (IBBTEC), Universidad de Cantabria-CSIC
c/Albert Einstein 22, PCTCAN
Santander, Cantabria E39011
Spain

Dear Dr. Lucas,

I am very pleased to accept your manuscript for publication in the next available issue of EMBO reports. Thank you for your contribution to our journal.

Kind regards,
